# Tree-based machine learning performed in-memory with memristive analog CAM

Giacomo Pedretti [1✉], Catherine E. Graves [1✉], Sergey Serebryakov[1], Ruibin Mao[2], Xia Sheng[1], Martin Foltin[1], Can Li [1,2] & John Paul Strachan [3,4✉]

Tree-based machine learning techniques, such as Decision Trees and Random Forests, are top performers in several domains as they do well with limited training datasets and offer improved interpretability compared to Deep Neural Networks (DNN). However, these models are difficult to optimize for fast inference at scale without accuracy loss in von Neumann architectures due to non-uniform memory access patterns. Recently, we proposed a novel analog content addressable memory (CAM) based on emerging memristor devices for fast look-up table operations. Here, we propose for the first time to use the analog CAM as an in-memory computational primitive to accelerate tree-based model inference. We demonstrate an efficient mapping algorithm leveraging the new analog CAM capabilities such that each root to leaf path of a Decision Tree is programmed into a row. This new in-memory compute concept for enables few-cycle model inference, dramatically increasing $10^3 \times$ the throughput over conventional approaches.

[1] Hewlett Packard Labs, Hewlett Packard Enterprise, Milpitas, CA, USA. [2] The University of Hong Kong, Hong Kong SAR, China. [3] Peter Grünberg Institute (PGI-14), Forschungszentrum Jülich GmbH, Jülich, Germany. [4] RWTH Aachen University, Aachen, Germany. ✉email: giacomo.pedretti@hpe.com; catherine.graves@hpe.com; j.strachan@fz-juelich.de

D eep neural networks (DNN) are becoming the mainstream model for numerous classification tasks such as image and voice recognition[1]. However, DNNs are unsuitable for multiple government[2] and industry[3] applications where inspectability and explainability are critical, training data may be limited, or where domain knowledge and historical expertise needs to be incorporated in critical decisions. These applications also include those in the medical space[4,5] where fast and accurate clinical assessments of a disease are critical as well as a deep understanding of the cause or reasons for a specific model classification result in order to rapidly prepare treatments. In these domains, tree-based methods, such as decision trees (DT) and their ensembles, for example random forest (RF) methods[6], are popular machine learning (ML) approaches due to their ease of training, good performance with small datasets[7] and reasonable interpretability for domain experts to verify and understand[8]. However, while fast to train, large-scale tree-based models are difficult to optimize for fast runtime (i.e., inference) without accuracy loss in von Neumann architectures[9]. In von Neumann architectures, storage and computing units are physically separated[10], which results in high energy consumption and time for data movement between the processor and storage[11]. Moreover, highly irregular memory access patterns to the model and feature vector for each DT node are nonuniform, and higher accuracy models require more and deeper DTs, resulting in unpredictable traversal times. State-of-the-art implementations run in super-linear time with DT depth, limiting scalability. Various approaches for speeding up RF[9,12–14] showed mainly incremental improvements as the data-locality access pattern problem remains.

A new class of accelerators where computation is performed inside the memory, termed in-memory computing (IMC)[15], is gaining momentum and accelerators for different applications such as neural network training and inference[16–19], image processing[20,21] and scientific computing[22–24] have demonstrated large performance improvements. Many such works utilize a core IMC primitive based on crossbar arrays of nonvolatile memory (NVM) devices[15], often dubbed memristors[25], to directly accelerate vector matrix multiplication. The IMC crossbar primitive combined with memristors, which can operate at low power and high speed[26], forms the basis of the dramatic performance improvements. However, implementation of tree-based ML algorithms in crosspoint arrays has not been shown yet, since these workloads are not dominated by matrix operations. Other traditional CMOS accelerator approaches have been studied for these models, such as an RF IMC accelerator based on complementary-metal-oxide-semiconductor (CMOS) static random-access memories (SRAM) in ref. [27], but model inference at high throughput and low energy operation remains a challenge.

Recently, another IMC primitive has been increasingly studied based on content addressable memories (CAM)[28]. CAM circuits natively perform a matching operation between an input data word (search key) and a stored set of data patterns in the CAM array in a highly parallel manner—thus accelerating a lookup operation. Traditional CAMs are based on SRAM, and show excellent throughput due to the parallel lookup at the cost of very large power consumption and area. Memristor-based CAM and ternary CAM (TCAM) circuits have been proposed[29–33] to produce lower-power CAMs, and their ability of performing high performance computation, such as finite automata inference[33–35] has been shown. While powerful, these approaches were limited to binary memristor states and did not take advantage of the analog continuous state tunability of memristive devices and the increased computational density. To fully leverage memristor capabilities, we recently developed an analog memristor-based CAM circuit and concept[36]. Instead of searching, storing and

outputting digital data, the analog CAM enables the search of analog values and the storage of analog ranges using the continuously tunable states of memristors, and compares an analog input with this stored range to determine a match or mismatch. This concept enables both a multi-bit encoding for each cell, or the ability to store continuous ranges. Other multi-bit CAM have also been proposed in ferroelectric technology[34].

Here we propose the first demonstration of accelerating the important class of tree-based ML models with an IMC approach utilizing the analog CAM. Our new concept efficiently maps root-to-leaf paths of tree-based models directly to analog CAM rows for few-cycle model inference, and is critically enabled by the unique features of the analog CAM, in particular the range storage and analog search, as well as compression from the 'X' or 'don't care' encoding, described below. To demonstrate this concept, we first present in detail the analog CAM circuit, an accurate behavioral model from a 180 nm taped-out hybrid CMOS-memristor chip[36] and detail how the tree-based models map to the analog CAM hardware. Then, by pairing analog CAM arrays with 1-transistor-1-resistor (1T1R) resistive random access memory (RRAM)[37] or memristor for majority voting, we show how to map RF models and also detail our hardware-aware compression techniques which leverage the unique features of the analog CAM to reduce memory utilization (i.e., area and power). Finally, we benchmark our approach against recent state-of-the-art approaches for RF model inference. Our in-memory computing approach with analog CAM outperforms the throughput of existing accelerators by 1000× with a 30× reduction of node energy to decision.

## Results

**Analog CAM compact model**. With analog CAM hardware, the highly irregular memory lookup patterns of tree-based machine learning models can be accelerated with IMC architectures, due to the analog CAM capability to store ranges of values and search analog data. Figure 1a shows the conceptual flowchart for implementing such models in CAMs, where after the generation of the ML model it can be compressed before deployment to optimize the performance. Figure 1b shows the working principle of a digital TCAM. Digital words are stored in different rows of the memory array. By applying a search word on the data line (DL), or columns, it is possible to rapidly search if the word is present in the memory, in which case the address is returned. Each match line (ML), or row, is initially precharged and remains charged only if all elements of the searched word match with that stored word. A wildcard 'X' representing an 'always match' is also possible to be stored or searched, allowing for a third (ternary) state in this TCAM. Different hardware implementations of TCAM have been proposed both based on traditional CMOS technology[28,38] and NVM technology[29–33]. To represent more than three levels in a TCAM cell one needs to be able to represent a range, defined by a lower and upper limit. For example, if the data $d = 13$ is to be represented, the memory cell should be able to accept any value $12.5 < i < 13.5$ where the 0.5 accounts for the system tolerance, namely half of the least significant bit or $LSB/2$. Figure 1c shows a conceptual schematic of an analog CAM array[36], where ranges are stored in memory, an analog word is given as input on the DL (columns) and the corresponding match is sensed on ML (rows) as a digital signal. In this case the equivalent of a wildcard 'X' corresponds to storing the range thresholds to the maximum acceptable limits, in this case 0 to 1, such that any input will match. Figure 1d illustrates a schematic for an analog CAM based on memristor technology[36], although other implementations, for example based on ferroelectric transistors[34] have recently been proposed as well. Range

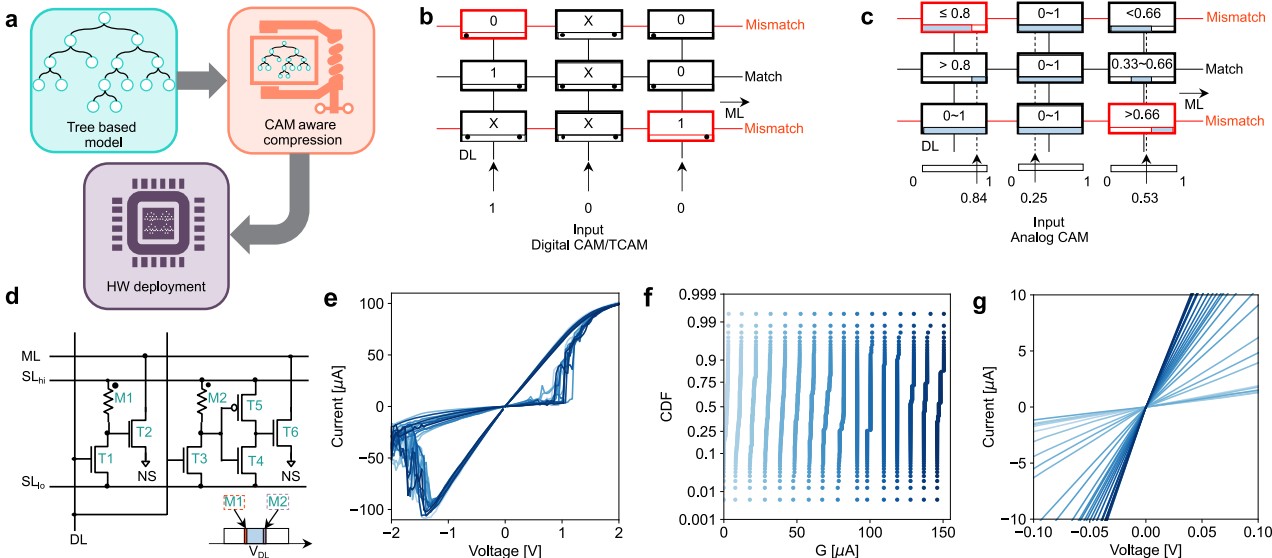

**Fig. 1 Analog content addressable memory with memristor. a** Illustration of this work, tree-based machine learning models are optimized and deployed on analog CAM hardware. **b** Digital ternary content addressable memory (TCAM), which searches a given input word across the whole memory and outputs the location of a match. **c** Analog CAM, where each cell stores ranges of values, or multi-bit representations. An analog input word is searched against the whole memory array in parallel, similar to the digital TCAM. **d** Circuit schematic of an analog CAM with memristor. **e** Memristor current–voltage (I–V) characteristics for different cycles. **f** Cumulative density functions of 16 levels of conductance corresponding to 4 bits of information, each programmed into 128 memristive devices. **g** I–V plot of read sweep for different conductances programmed in memristors showing linear behavior.

thresholds are stored in the memory conductance M1 (lower threshold) and M2 (upper threshold) as shown in the inset. By applying a DL analog value, a voltage divider between memory device and the series transistors T1 and T3 controls the discharge transistor (T2) on the lower threshold side, or the inverter (T4–T5) on the upper threshold side which controls the upper threshold discharge transistor (T6). Note that not only multi-bit values but also ranges can be stored through the programming of the left and right conductances.

The lower and the higher bound of the searching range is stored as conductance in the RRAM device in our analog CAM. The RRAM device current–voltage (I–V) characteristics are shown in Fig. 1e, with the device structure fabricated in a back-end-of-line (BEOL) process illustrated in Supplementary Information Fig. 1. A TaOx dielectric layer is sandwiched between metallic top electrode (TE) and bottom electrode (BE) and the structure is realized on a conventional 180 nm complementary-metal-oxide-semiconductor (CMOS) process[36,37,39] (see Methods). A newborn device is typically in a high resistance state due to limited conduction in the dielectric layer. After a forming procedure, oxygen vacancies are reordered such that a conductive path is formed between TE and BE resulting in a low resistance state (LRS). Then by applying a negative reset pulse, the conductive path can be retracted and the device results in a high resistance state (HRS). Conductance can be modulated from LRS to HRS by switching positive and negative voltage. Moreover, switching to different intermediate states can be controlled through a variety of means, including through current compliance ($I_C$) modulation, namely the maximum current flowing into the RRAM device during the set transition controlled with a series transistor, or by means of $V_{stop}$, or the maximum voltage applied during the reset operation[26,40]. Figure 1f shows the cumulative distribution function of 16 different levels measured on 2048 devices on a large array[37] (Supplementary Information Fig. 2), corresponding to 4 bits, demonstrating the possibility of analog and multi-bit capability. If a larger number of bits is needed, multiple cells can be used in parallel, with a bit-slicing technique,

similar to what is typically done in crosspoint arrays[41]. For small applied voltages, memristor devices offer a linear conduction as shown in Fig. 1f, especially for states close to LRS levels. The linear dependence simplifies the interpretation of the voltage divider in the analog CAM circuit.

We taped-out an analog CAM array in 180 nm CMOS technology[36], with the cell layout shown in Fig. 2a. We also carried out extensive circuit simulations on a more aggressive 16 nm technology node[36] to study the impact of power consumption and scalability of the design. However, to have a fast deployment and performance assessment of more complex and large-scale problems, such as DT/RF and other tree-based machine learning algorithms, a compact and reliable analog CAM cell model based on the actual analog CAM taped-out should be realized. SPICE circuit simulations can be computationally expensive for large-scale systems, compared to small-scale arrays, and may not comprehensively include true process variations and parasitics. Thus they can be improved with data from taped-out chips. For this reason we designed a compact cell model whose details are illustrated in Supplementary Note 1 and Methods. Figure 2b shows circuit simulation results of the current flowing in the lower threshold branch, or in transistor T2, as a function of $V_{DL}$ and M1 programmed conductance $G_{M1}$, where a large current corresponds to a low $V_{DL}$ or high $G_{M1}$ as expected. Figure 2c shows the model calculation for the same current, which is in good agreement with the circuit simulation. Figure 2d-e shows the circuit simulation and model calculation, respectively, of the current flowing into the upper threshold branch, or in transistor T6, where in this case high current corresponds to high $V_{DL}$ or low $G_{M2}$. Data and calculation are in good agreement, confirming the model reliability. Note that circuit simulations were performed by taking into account post-tape-out parasitic effects (see Methods), thus the model comprehensively describes the cell behavior in a reliable manner. Figure 2f shows data (circle) and model calculation (lines) of two different ranges programmed in two analog CAM cell[36], which confirms that the model can accurately predict cell behavior.

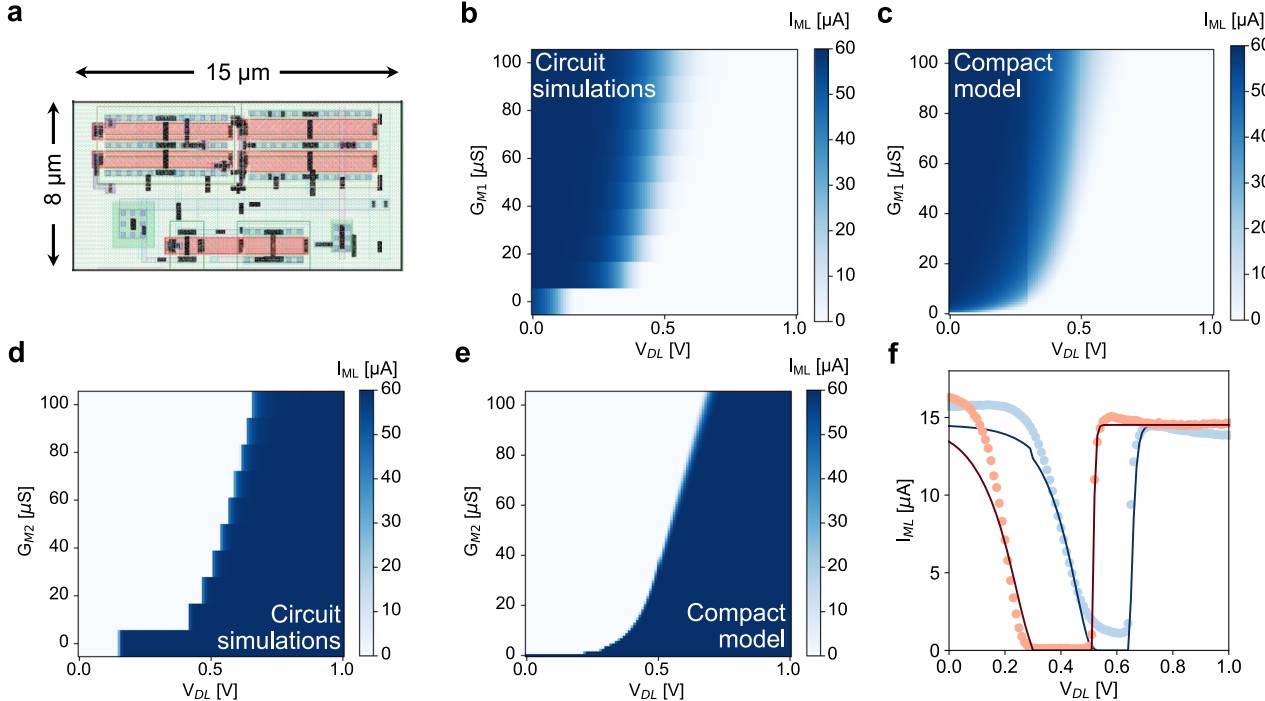

**Fig. 2 Analog CAM compact model. a** 180 nm cell layout showing the various inputs/outputs. **b, c** Circuit simulation and model calculation of ML discharge current on the lower threshold branch as a function of $V_{DL}$ and $G_{M1}$. **d, e** Circuit simulation and model calculation of ML discharge current on the upper threshold branch. **f** Experimental data (circle) and compact model (lines) for two different ranges stored in analog CAM cells.

**Mapping DT/RF to analog CAM**. DT are powerful ML models allowing data classification and regression, with much clearer understanding of the resulting models than deep learning techniques. As a toy example, Fig. 3a shows a DT trained to classify the Iris dataset[42], where features namely sepal and petal width and length are organized as a feature vector $f = [f_0, f_1, f_2, f_3]$ and given as input. At each node a decision on a feature is made according to a threshold. If the decision is positive, the tree is traversed from top to bottom following the left branch; if the decision is negative, the right branches are taken. Trees are traversed until reaching a leaf, which in this case corresponds to the classes of Iris, namely Setosa, Virginica or Versicolor. DT can be mapped to analog CAM arrays by directly programming each root-to-leaf path into an array row. Feature vectors $f$ are given as input to the columns DL, and ML is initially precharged. If all the analog CAM cells of a row match $f$, ML stays charged—otherwise ML discharges into the unmatched analog CAM cell. Note that this corresponds of doing an AND operation between every analog CAM cell in a row. Figure 3b shows the implementation of the DT of Fig. 3a into an analog CAM array. If a feature component is not present in the root-to-leaf path, a wildcard 'X' can be inserted corresponding to the whole range programmed in the analog CAM, i.e., the LRS on the lower threshold memristor and the HRS on the upper threshold memristor, as can be seen in the $f_1$ column. If a feature is present multiple times in a branch and with different thresholds, for example in the second row, third column, then the two thresholds are combined and a range is encoded. In the case of only one threshold decision for a particular feature, one of the memristors is kept as a wildcard (LRS or HRS) and the other is programmed at an intermediate threshold value, implementing a 'less than or equal to' with a high threshold (a left branch), while a 'greater than' is programmed in the opposite case (a right branch). While the presence of a large number of 'X' appears as a drawback it will be actually used for allowing compression as it will be shown in the next section. MLs of the matching rows directly correspond to the matching class,

making an analog CAM able to perform a one-step classification independent of the array size, corresponding to three clocks cycles $t_{CLK}$ for charging ML, asserting DLs, and latching MLs after $t_{CLK}$. Note that not only DT can be mapped to analog CAM but also in principle any tree-based ML algorithm comprising root-to-leaf decision paths. Multiple types of trees can be mapped to CAMs independent of the tree hyperparameters such as depth or height. When the size of the problem exceeds the maximum size of an array, multiple arrays can be used in a tiled architecture as described in the following section. To verify that analog CAM arrays are effectively able to draw decision boundaries we trained 12 different DTs with all possible intersections of two features. We observe in a two-dimensional space the classification results by deploying the DTs on analog CAM arrays using the compact model and monitoring the matched rows. Figure 3c shows a plot of the ML predicted class (shadows) and ground truth results (circles) as a function of different DL voltages corresponding to different feature vector values. Circles landing on a shadow with matching color corresponds to a correct prediction. The trend suggests that the DT model deployed on analog CAM draws the correct decision boundaries in the classification task. Figure 3d shows a plot of the ML discharge currents (left) in each analog CAM cell of Fig. 3b as a function of $V_{DL}$ and the corresponding ML outputs as a function of time for the same feature vector of Fig. 3a-b, demonstrating the ability to recognize the correct class. The corresponding conductance values mapped in the analog CAM array are shown in Supplementary Information 3. This example corresponds to the first demonstration of mapping DT root-to-leaf path with IMC due to analog CAM. With the wildcard and range encoding capabilities, analog CAM is beautifully well suited for mapping such computations, which is complex to accelerate in conventional and custom hardware.

While DT are easy to train and deploy, their accuracy for real world problems is affected by overfitting. This is due to the need for more tree depth to effectively minimize the cost function during training. To avoid this, ensemble methods are used in which

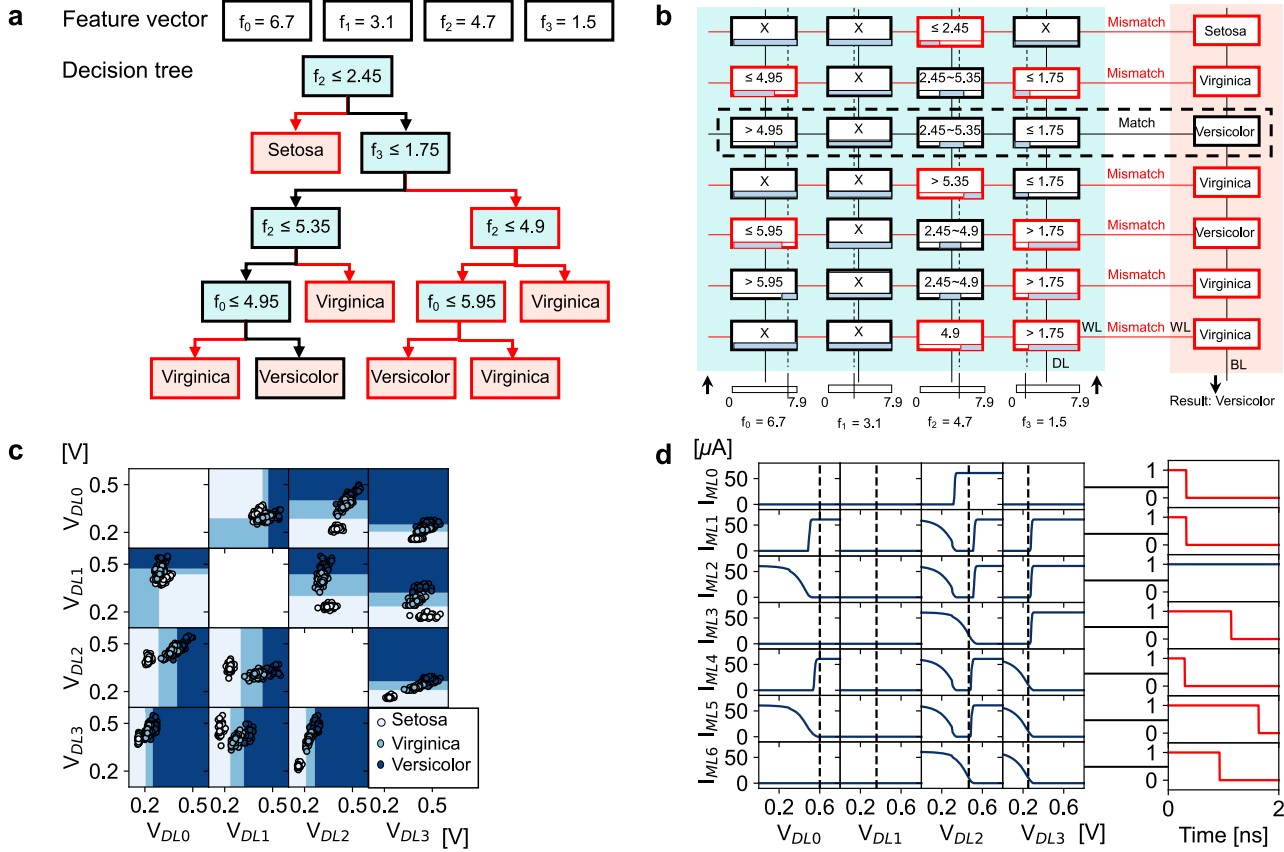

**Fig. 3 Mapping decision tree on analog CAM. a** Decision tree for classifying the Iris dataset; a feature vector is given as input and features are compared with learned thresholds to decide which branch of the tree to take. **b** Analog CAM array mapping the decision tree of **a**, where each root-to-leaf path is written in an array row. Classification outputs are given on the ML, in one shot. **c** Decision map of different DT trained with two features of the feature vector, calculated in the analog CAM. **d** Current flowing in every analog CAM cell as a function of $V_{DL}$ and resulting ML digital output for the inference of feature vector and DT in **a**.

multiple trees are evaluated in parallel. This is the case for RF, consisting of $n_{trees}$ inferred in parallel, with the output result computed as the majority gate of single DT outputs. Each DT in a forest is trained with a small random portion of the dataset and usually requires only a shallow depth and relatively small number $n_{trees}$ to reach good accuracy. Both tree inference and a majority vote can be implemented with IMC. Figure 4a shows an architectural overview of the RF inference acceleration approach. Inputs are applied on the DL with a digital to analog converter (DAC, see Supplementary Information Fig. 7). Each root-to-leaf path of each DT is mapped to a row of the analog CAM array, whose ML outputs are converted to a digital high or low signal with a sense amplifier, whose circuit is illustrated in Supplementary Information Fig. 8. Sense amplifier outputs are connected to the gate of a one-transistor-one-resistor (1T1R) memristor (RRAM) array[37], with every column sensing a corresponding class. The 1T1R RRAM array $M$ is programmed such that $M[i, j] = LRS$ if class($i$) = class($j$), where $i$ corresponds to the ML index and $j$ the column index, and columns correspond to a different class. In this way, the more ML is activated of a given class, the larger the current flowing into the 1T1R RRAM array column for that class. Currents are sensed with a typical chain consisting of a trans-impedance amplifier (TIA), sample and hold (S&H), and analog to digital converter (ADC)[37]. Note that only 4 clock cycles, corresponding to precharging the ML, asserting DL, evaluating the root-to-leaf path with the SA latch, and triggering the RRAM read, are needed to reach a classification result and, as a first approximation, this is independent of the number of trees in the forest.

To test our system and compare results with previous benchmarks[27], we implemented an RF for the classification of KUL Belgium traffic sign dataset[43] whose data processing is explained in Supplementary Information 4. We mapped the RF into the analog CAM and RRAM arrays and evaluated the accuracy of inference on 200 samples[27] reaching 0.965, higher than the reference state of the art. These RF models are well matched to analog IMC implementations, showing strong resilience to variation and noise that can otherwise affect analog hardware. As an example, Fig. 4b shows the RF accuracy as a function of the standard deviation of a Gaussian distribution representing the variability in the memristor conductances, which captures a practical challenge in some memristive devices[40]. Accuracy remains unaltered for a standard deviation up to $\sigma_G = 5\%$, which can be realized in practical and size-scaled devices[39]. Figure 4c shows accuracy loss as a function of the number of bits considered in programming the analog CAM threshold demonstrating that only for fairly low bit numbers, i.e., $N_{bit} < 3$ the accuracy degrades considerably.

**Architecture optimization**. To directly deploy our RF model to an analog CAM, a very large array, i.e., $2000 \times 256 = 512$ kb, is needed for one-shot classification. However, most of the analog CAM cells in an RF implementation remain empty. In fact, each root-to-leaf path has a maximum size of the number of decision nodes, a hyperparameter that can be defined during training and known as maximum depth. Typically the maximum depth is

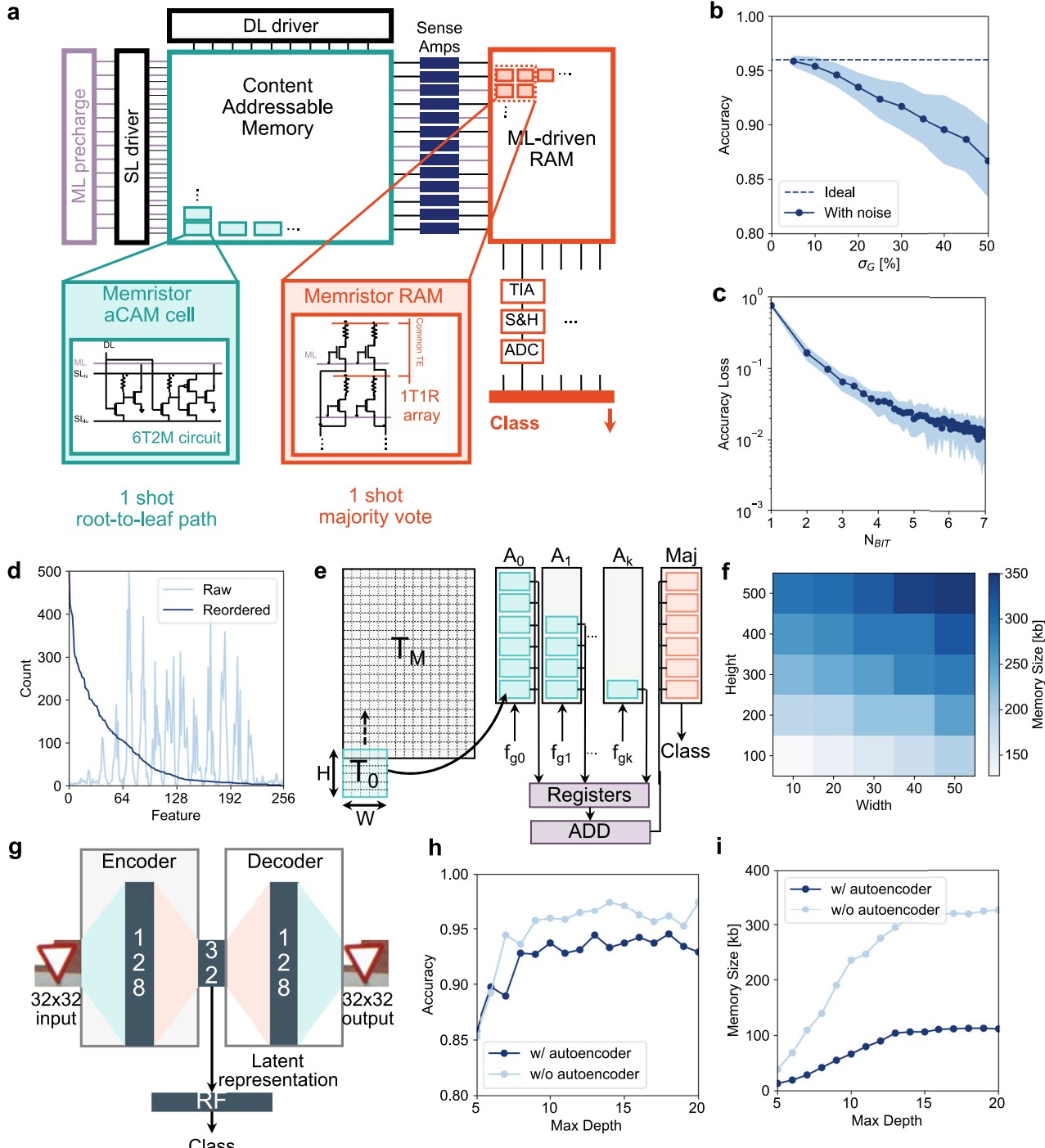

**Fig. 4 RF accelerator architecture. a** Overview of the full IMC system, with analog CAM executing root-to-leaf evaluation in one step and 1T1R RRAM array executing the majority vote in the analog domain. **b** Accuracy as a function of the standard deviation of injected noise $\sigma_G$ in the programmed conductance for 100 different experiments, filled line represent the average while bands correspond to the standard deviation. **c** Accuracy loss as a function of the number of bit for representing the threshold in 10 RF trained on the dataset, filled line represent the average while band the standard deviation. **d** The number of populated (non `X`) analog CAM cells in the arrays as a function of the feature index for a direct mapping (raw, light blue line) and reordered array (blue line). **e** Tiles and array mapping procedure, the threshold map $TH_{Map}$ traversed from bottom to top and from left to right by $H \times W$ tiles ($T_0$ in the example), which are filled in the presence of valid rows, namely rows that are not completely empty. Once a tile is full, it is placed in the corresponding array ($A_0$) in the example, which will evaluate a group of $W$ features from the feature vector. **f** Memory size needed as a function of tile height $H$ and width $W$. **g** Schematic of variational autoencoder architecture for compressing the dataset a 32 wide feature vector. **h** Accuracy as a function of maximum depth in the RF with and without autoencoder as input. **i** Memory size required as a function of maximum depth in the RF with and without autoencoder as input.

small compared to the number of available features; in the present reference dataset the feature vector length is $F = 16 \times 16 = 256$, and good RF accuracy is reached with a maximum depth ~10 (Supplementary Information Fig. 5). Figure 4d shows the number of occupied cells in the analog CAM array as a function of the feature identifier. As seen from the 'Raw' (light blue) line, some features occur frequently and correspond to 'important' pixels in the training images (for example defining the shape of the traffic sign), but other features are hardly considered (for example the border of the images) and only a few analog CAM cells of the corresponding columns are occupied. Given that the feature order is arbitrary in each individual root-to-leaf path, we reordered them based on occurrence such that all-important features are on the left side of the entire analog CAM array. In this way part of the array remains completely off, or empty. Moreover, we similarly reorder the columns to make sure that the most populated columns are on the bottom of the analog CAM array. The blue line of Fig. 4d shows the reordered count, demonstrating that a part of the array can remain empty and offering compressibility once the large RF array is tiled onto reasonable-sized analog CAM arrays.

We investigated efficient architectures for mapping a large RF model by exploring CAM array tile sizes and available compression schemes. We divided the CAM architecture into tiles of practical size $H \times W$, i.e., up to $480 \times 48$, dimensions that were previously found to be feasible[33,36]. Figure 4e shows the tile writing procedure and the tiled architecture. Given a target threshold map $TH_{Map}$ following the reordering procedure we start by sweeping an $H \times W$ tile $T_0$ on the left part of the array, i.e., we evaluate $TH_{Map}[0, 0 : W]$ and if there is at least one cell not empty we accept the row and write it in $T_0[0, 0 : W]$, otherwise we discard it. We continue evaluating $TH_{Map}[i, 0 : W]$, with $i = 1$ and writing in $T_0[j, 0 : W]$ incrementing $i$ at every cycle and $j$ only if the location is written until $T_0$ is filled. We proceed by positioning $T_0$ at array location $A_0$, which will evaluate the feature group $f_{g0}$ corresponding to the first $W$ features. We take a new tile $T_1$ and start filling it and repeat the process until all elements of $TH_{Map}[:, 0 : W]$ have been considered. Once this part of $TH_{Map}$ has been mapped, we increment the column, namely, we start evaluating $TH_{Map}[0, W : 2W]$, and place the corresponding tiles in array location $A_1$, which evaluates feature group $f_{g1}$ corresponding to features $W \sim 2W$. The process is repeated until all of $TH_{Map}$ has been evaluated and all the $k = F/W$ arrays are populated. Note that most tiles of $A_0$ are populated while most of $A_k$ is empty, thanks to the reordering. In this way, most of the right-side arrays tiles can be eliminated. The output of each tile is collected in a register and logically added to perform the final majority vote only one time in an RRAM memory. Figure 4f shows the CAM memory size as a function of the tile dimensions $H$ and $W$ after reordering and mapping, demonstrating significant compression compared to the initial size of 512 kb. We finally choose the training hyperparameters for benchmarking our system to yield good accuracy with a reduced memory size (Supplementary Information Fig. 5), namely an RF with 15 trees and a maximum depth of 10 is chosen here.

The memory size can also be compressed by reducing the dimension of the input images, either by preprocessing the data with principal component analysis (PCA), independent component analysis (ICA), or with an autoencoder. Figure 4g shows a standard variational autoencoder schematic, with 128 hidden neurons in the decoder and encoder path and a latent space of 32 elements. We trained the RF with the dataset preprocessed with the autoencoder, namely with a feature vector size of 32 (Supplementary Information Fig. 6). Figure 4h shows the classification accuracy as a function of the maximum depth of the trees with and without autoencoder preprocessing, where it is

possible to obtain a loss of a few percent in exchange for significant compression. Figure 4i shows the memory size using $32 \times 32$ tiles, for encoding the RF with and without autoencoder, demonstrating a compression factor close to the latent space dimension compared with the original dimension.

**Performance evaluation.** To evaluate the power consumption and throughput of the analog CAM system, we considered a full circuit including an ML precharge circuit, sense amplifier, a digital to analog converter[36] for charging the DL (Supplementary Information Figs. 7 and 8), and memristor conductances from the data of Fig. 1e. For each decision boundary, the corresponding conductance to the map was directly extracted from the distribution such that hardware statistical variations are taken into account. To study the design hyperparameters namely tile size $H$ and $W$, and clock frequency $t_{CLK}$ we evaluate the accuracy, throughput, power consumption, and energy per node per decision, namely the energy spent for assessing each threshold in a tree. Figure 5a shows accuracy as a function of $t_{CLK}$ for different $H$ and a fixed $W = 16$. As expected, accuracy does not depend on $H$ but there is a dependence on $t_{CLK}$, as enough time should be given to ML for discharging if the input does not correspond to a match. However, good accuracy is preserved for $t_{CLK} > 1ns$, which guarantees a high throughput up to $60 \times 10^6$ Decisions/sec, as shown in Fig. 5b. Note that the throughput does depend on $W$, in fact, arrays $A_i$ are evaluated one by one, thus a smaller tile size $W$ corresponds to a larger number of arrays $A$ and latency. However, with a continuous input flow, operations can be pipelined by for example charging ML of $A_1$ while latching the ML result of $A_0$ and throughput can be highly increased at the cost of power consumption, at fixed energy per decision. Figure 5c shows the dynamic power needed for charging and discharging the ML as a function of $t_{CLK}$ for different $W$, demonstrating a dependence on $W$ due mostly to tiling. ML power is dominated by the precharge and sense amplifier circuit but also the memory contribution becomes significant for $W > 10$. In Fig. 5d it is reported the dynamic power consumption needed for charging the DL, considering the proposed DAC and an optimum $R_{out}$ (Supplementary Information 7). While dynamic power consumption is low, static power consumption due to the voltage divider evaluation of M1-T1 and M2-T3, which is shown in Fig. 5e as a function of $H$ and $W$ is quite important and should be carefully taken into account while choosing the hyperparameters. Finally, Fig. 5f shows the energy per node per decision as a function of $H$ and $W$ at $t_{CLK} = 1ns$. Finally, Supplementary Information Fig. 9 shows the energy per decision as a function of $H$ and $W$ for the pipelined architecture, assuming a constant input data stream, which has an opposite trend compared with Fig. 5d. Taking into account this dependence we chose tiles of size $16 \times 480$ and maximize the performance of the pipelined architecture, leading to a total number of 29 needed arrays for mapping the problem. Supplementary Information Fig. 10 shows a breakdown of the energy consumption for the various components.

Finally, we compared the performance of our approach to existing ASIC and conventional proposals. For a fair comparison to the ASIC works, we evaluated our analog CAM hardware on a 65 nm technology by applying a constant field scaling procedure. We considered a maximum clock frequency of 1 GHz, as a practical case[27], and evaluated the performance of our architecture compared with different results from literature[12,14,27,44]. The comparison is shown in Table 1 with analog CAM outperforming existing accelerators in throughput and energy per decision[27]. Moreover, the algorithm independent metrics, normalized by the number of nodes of each tree once again shows the strong

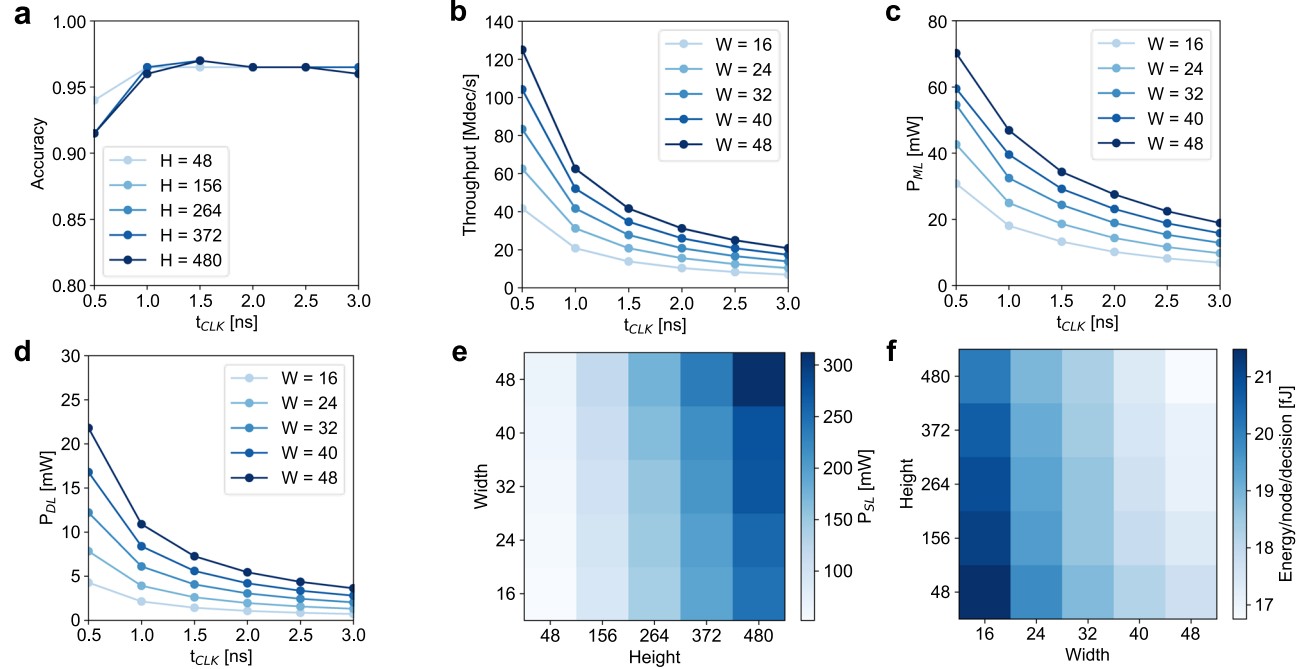

**Fig. 5 Performance evaluation. a** Classification accuracy as a function of $t_{clk}$ for different tile height $H$ at fixed tile width $W = 16$. Accuracy does not depend on $H$ as expected, until for nominal operational frequency namely for $t_{clk} > 1ns$, below which mapping in the chosen conductance range is not possible. Throughput (**b**), ML dynamic power consumption (**c**) and DL dynamic power consumption (**d**) as a function of $t_{clk}$ for different $W$ at fixed $H = 48$. **e** SL static power consumption as a function of $H$ and $w$ for $t_{clk} = 1ns$. **f** Energy per DT node per decision as a function of $H$ and $w$ for $t_{clk} = 1ns$.

**Table 1 Comparison of tree-based ML accelerators in literature with this work.**

| Accelerator | Process | $f_{clk}$ | Power | Throughput | Energy | Node energy | EDP | Node EDP |
|---|---|---|---|---|---|---|---|---|
| | [nm] | [Ghz] | [mW] | [Dec/s] | [nJ/dec] | [pJ] | [aJs] | [aJs] |
| Intel X5560[44] | 45 | 2.8 | $190 \times 10^3$ | $9.3 \times 10^3$ | $20.4 \times 10^6$ | N/A | N/A | N/A |
| Nvidia Tesla M2050[44] | 40 | 2.8 | $225 \times 10^3$ | $20.4 \times 10^3$ | $11 \times 10^6$ | N/A | N/A | N/A |
| Xilinx Virtex-6[44] | 40 | 0.079 | $11 \times 10^3$ | $31.3 \times 10^3$ | $351 \times 10^3$ | N/A | N/A | N/A |
| ASIC[12] | 65 | 0.2 | 5.6 | 30 | $186.7 \times 10^3$ | 22 | $6.2 \times 10^{12}$ | $740 \times 10^3$ |
| ASIC[14] | 65 | 0.25 | 27.6 | 60 | $460 \times 10^3$ | 1.4 | $7.7 \times 10^{12}$ | $24 \times 10^3$ |
| ASIC IMC[27] | 65 | 1 | 7.1 | $364.4 \times 10^3$ | 19.4 | 9.8 | $53.2 \times 10^3$ | 27 |
| This work | 65 | 1 | 26.74 | $20.83 \times 10^6$ | 1.28 | 0.32 | 61 | $15 \times 10^{-3}$ |
| This work pipelined | 65 | 1 | 427 | $333 \times 10^6$ | 1.28 | 0.32 | 3.84 | $0.9 \times 10^{-3}$ |

improvement of accelerating RF with an analog CAM architecture compared to other works, with an energy-delay product per node 3 orders of magnitude lower the state of the art (see Supplementary Information note 2). We note as well that our analog CAM approach is flexible and can be optimized for specific applications—for example, by using a pipelined architecture one can modulate the amount of power spent at the cost of a reduced throughput or vice-versa, in a constant-energy power/accuracy trade-off. We also compare the area previously shown for the layout of analog CAM at 16 nm technology node[36], with a scaled SRAM-based implementation at the same technology node considering all peripheral involved (see Supplementary Information node 3). While the area of 6T2M analog CAM is ~2× larger than the 6T SRAM, the area efficiency of analog CAM implementation is ~142× larger. In fact, with the nonvolatile and analog behavior of memristor, which represent the decision boundaries in analog CAM, the computational density reaches unprecedented peaks.

We envision that such results open the possibility for analog CAM to accelerate different tree-based workloads, including state-of-the-art AI tasks that usually require high energy consumption for training and inference[45].

## Discussion

In summary, we have proposed a tree-based machine learning accelerator with IMC primitives based on analog CAM, in which by mapping root-to-leaf paths to CAM array rows it is possible to perform rapid parallel inference. A post-layout compact model of the analog CAM was designed to assess performance on RF inference as part of a larger CAM-RRAM system implementation. Results at a scaled technology node demonstrate up to ~$10^3$× higher throughput and ~12× reduced energy per decision compared with the state-of-the-art, resulting in >$10^4$× lower EDP. Our work lays the foundation for novel accelerators based on analog CAM as a radical new computing primitive side-by-side with crosspoint arrays. The high performance offered for this class of machine learning models possessing increased explainability provides a compelling opportunity to use analog CAM-based hardware in critical application areas.

## Methods

**Memristor integration.** The memristors were monolithically integrated on CMOS fabricated in a 180 nm technology node. The integration starts with a removal of silicon nitride and oxide passivation with reactive ion etching (RIE) and a buffered oxide etch (BOE) dip. Chromium and platinum bottom electrodes are then patterned with e-beam lithography and metal lift-off process, followed by reactive sputtered 4.5 nm tantalum oxide as switching layer. The device stack is finalized by e-beam lithography patterning of sputtered tantalum and platinum metal as top electrodes.

**Analog CAM circuit simulation.** 6T2M analog CAM cell and small arrays were designed and simulated in Cadence Virtuoso Custom IC design environment, and the simulation result post-processed with HP-SPICE. The simulations utilize the TSMC 180 nm and 16 nm library and the designs follow the corresponding rules. A custom python script generates the netlist for analog CAM arrays with different numbers of rows and columns and arbitrary configured memristor conductance and input voltages.

**Analog CAM model.** Analog CAM model was implemented in Python environment by fitting outputs of circuit simulation with simplified physical laws and behavioral equation. While mostly in the subthreshold regime, input transistor T1(T3) was modeled both in subthreshold and ohmic conduction regimes. The first obey the simplified MOS model:

$$I_{D1} = I_{D0} \exp\left(\frac{V_{DL}}{\alpha}\right) \quad (1)$$

with $I_{D0}$ and $\alpha$ fitting parameters (for details see Supplementary Information 1). Given that T1 drain-source voltage (corresponding to the voltage divider) $V_{div} = V_{G,T2} - V_{SL,lo}$ is typically low, or $V_{div} < V_{GS} - V_T$ with $V_{GS} = V_{DL} - V_{SL,lo}$ and $V_T$ threshold of T1, in the region of interest, we assume that it can only be either in subthreshold or ohmic (linear) region, whose current obeys to the law:

$$I_{D1} = k_1(V_{GS} - V_T) \quad (2)$$

with $k_1$ fitting parameter corresponding to physical and electrical properties. Once the voltage divider has been computed, the ML discharge current can be computed as:

$$I_{D2} = k_2(V_{div} - V_T)^2 \quad (3)$$

Being the output transistor T2 (T6) is typically biased in the saturation region, at least in the initial discharging phase. Finally, the inverter was modeled as a sigmoid for simplicity and fast calculation:

$$V_{G.T6} = \frac{-0.8}{1 + \exp(-\beta(V_{div} + \gamma))} + 0.8. \quad (4)$$

Parastic parameters were extracted from the post-layout simulation and correspond to a resistance connecting each cell namely $R_{ML} = R_{DL} = r = 1.4\Omega$ and a parasitic capacitance of $C_{ML} = C_{DL} = c = 1.9fF$. Precharge block and sense amplifier were assumed as parasitic capacitance, which was extracted from the post-layout simulation as $C_{PC} = 40.95fF$ and $C_{SA} = 50fF$, respectively.

**Models training.** All tree-based models were trained in a Python environment with sklearn module. To match the benchmark, we trained RF with KUL Belgium traffic signs dataset[43] considering the same 8 class and training/testing set as in literature[27], thus we used 2300 training and 200 testing images taken from the classes 'No Overtaking', 'Children', 'Crossroads with a minor road', 'Priority road', 'Give Way', Stop', 'No vehicles', and 'Maximum speed limit'. However, while the reference RF was trained with 64 trees and a maximum depth of 6, we optimized the hyperparameters namely maximum depth and number of trees reaching an accuracy of 96.5% when deployed to analog CAM, overcoming the given accuracy of 94%.

**Power consumption calculation.** Power consumption calculation of the pipelined architecture was divided in three parts, namely

- static power consumption flowing into the voltage divider

$$P_{\text{static}} = V_{sl,hi}I_{D0} \quad (5)$$

- dynamic power consumption to charge the DL

$$P_{DL} = \frac{V_{DD}^2 WN}{R} \quad (6)$$

  with $W$ tile width, $N$ number of tiles, and $R$ output resistance of the DAC (Supplementary Information)

- dynamic power consumption to charge and discharge the ML

$$P_{ML} = \frac{1}{2t_{CLK}}(C_{ML}V_{ML0})^2 HN + \sum_{j=0}^{i=N}\sum_{i=0}^{i=H}\frac{1}{2t_{CLK}}(C_{ML}(V_{ML0} - V_{ML,i,j}))^2 \quad (7)$$

with $V_{ML0}$ initial voltage of the ML, which can be set from the precharge block, $H$ tile height, and $V_{ML,i,j}$ ML voltage of row $i$ of tile $j$ at $t = t_{CLK}$. The first term corresponds to the charging energy and the second to the discharging in each cell.

## Data availability

The data that support the findings of this study are available from the corresponding author upon reasonable request.

## Code availability

The code used to generate the results of this study is proprietary to Hewlett Packard Enterprise.

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

## Author contributions

G.P., C.G., C.L., and J.P.S. conceived the tree mapping procedure, G.P. designed the compact model, C.L., X.S., and R.M. conducted experiment, G.P. and C.L. analyzed the data, G.P. and S.S. designed the machine learning models, G.P., C.G., and J.P.S. conceived the compressing procedure and architecture, G.P. and M.F. performed benchmark and scaling calculations. All authors reviewed the manuscript. J.P.S. supervised the research.

## Competing interests

The authors declare no competing interests.
