## [Peer Review File · Nature Communications]

Reviewers' Comments:

Reviewer #1:

Remarks to the Author:

The paper proposes the design of accelerators for Random Forest (RF) algorithms using their previously proposed (Ref. 36) analog memristor-based content addressable memory (CAM). Large RF inference models are mapped to arrays of analog CAM. Measured data from Ref. 36 is used to calibrate the CAM cell model in 180nm CMOS. Then, using circuit models of the 180nm CAM cell as well as extrapolated models in 16nm CMOS, the authors demonstrate significant gains in throughput (3-orders-of-magnitude) and energy-efficiency (2-orders-of-magnitude) gains over previously proposed works. These results are shown in simulations.

Specific comments are provided below:

1. It is refreshing to see a focus on something other than deep neural networks (DNNs). RF are powerful ML models but haven't been studied from the perspective of implementation. Furthermore, such models have the advantage of robustness and interpretability as is mentioned by the authors. On a related note, the manuscript mentions RF and DT as two distinct ML models when in fact RF is an ensemble of DTs. It would be better to drop the mention of DT unless it is used as a RF component.
2. A major issue with this paper is the reliance on simulations using circuit models to prove the benefits of the proposed architecture. While there is nothing wrong with the approach, in my opinion, this type of work is better suited for other journals in the devices area for example IEEE JxCDC. Nature papers need to represent a major breakthrough of sorts involving basic sciences. It was difficult to find any such breakthrough in this paper even though the material itself was interesting.
3. Related to Comment 2, the comparisons in Table 1 are highly unfair. For one, this table compares ASIC implementations in Refs. 13, 15, and 26 with simulations results using circuit models of the CAM cell. A well-known truism is that real-life implementations don't measure up to simulations in terms of the results. The work would be impactful if the authors were to build a complete CAM array including all peripherals and then report results. In that case, even a 4X-10X gains in energy efficiency would be very impressive provided these are done at iso-accuracy.
4. Related to Comment 3, comparison with Ref. 15 is very unfair since it is a digital processor for video object recognition at 60 fps. Data volumes needed to execute such computations is much larger than classification task on the KUL Belgium data set using RF as done in the paper. Similarly, fully programmable processors such as Intel (Ref. 45) or Tesla (Ref. 45) or Virtex (ref. 45) is very easy to beat in performance and power. Comparison with Ref. 26 would have been reasonable but, as mentioned in Comment 3, this is a real-life IC with measured results. Energy and throughput need to be compared for the same task (classification) and using the same inference model and at the same accuracy.
5. The paper is missing comparison using arithmetic efficiency metrics such as TOPS (tera operations per second) and TOPS/W. Another important metric is computational density in TOPS/mm². This is a specially important metric since one of the purported advantages of RRAMs is their density.
6. The proposed CAM cell is a 6T-2M cell with two resistive devices and 6 NMOS transistors. Transistors T1 and T3 along with their memristive loads act as threshold gates that turn ON at different values of DL. The rest of the functionality is implemented using NMOS transistors. This role of the memristive device seems not as central as it is in an architecture such as a cross-point. Both the area and power seem to be dominated by the CMOS switches. If Figure 2a were to mark the RRAM devices, this issue would be clear. Is this a good use of memristive devices? One benefit of NVM devices is the increase in computational density. The proposed CAM cell seems to miss out on this opportunity.
7. The manuscript is well-organized overall. However, it has a number of awkwardly put

sentences, e.g., second sentence on page 1: "However, DNN impact is limited for a range of applications where...." when the authors mean to say that "DNNs are unsuitable for applications where explainability is critical". Similar sentences are interspersed throughout. The authors are urged to revise such statements.

8. Typos – page 1, second para, line 8: "...is used in analog..." needs to be "...are used in analog..." since the reference is being made to "(NVM) devices,...".

In summary, while this is a worthwhile effort, the manuscript is not ready for prime time yet. Experimental results and fair benchmarking is required.

Reviewer #2:

Remarks to the Author:

This paper proposes to use NV ACAMs to implement DT/RF algorithms in hardware. Here are my major (MA)/minor (MI) comments:

(MA-1) How much are the DT/RF algorithms still useful and practical?

We may find more recent/efficient algorithms. The references that the authors have cited do not convince me. Citing Ref.2 and Ref.3 surprises me. Ref. 4 is a survey paper and has cited two (2016/2017) papers for DT algorithms. I could not access ref.5. Ref.s 6-8 are pretty old, and the only prestigious/recent reference about the importance of DT/RF algorithms is ref.9.

(MA-2) How much is an ACAM array flexible for various DTs? Each tree can have different shapes and different numbers of nodes.

(MA-3) I am not sure about the practicality of an ACAM array to realize a DT algorithm.

(1) Data input can be a voltage value between 0 to 1V or 0 to 1000mv. 0-1V is very limited. 0-1000mv might not be that practical to cover any value range for a DT. e.g., what if we need to check $523\text{mV} < \text{data} < 789\text{mV}$? How can we program the memristors to give us that accurate range? If a degree of inaccuracies is acceptable by the algorithms, it should be discussed/evaluated in the paper.

(2) How to store an 'X' state to an MCAM cell? I think (not sure) the lower band cannot be as low as 0V.

(3) Having a rather large row, let's say a 64-cell row, how much the array can detect the worst-case mismatch cases? e.g., we have 63 matched cells, and we do have one mismatch cell while the input query is very close to the lower/upper band. Can the row detect this marginal case as a mismatch case?

(4) What about the worst-case match cases? We have 64 cells, which all are inside the matching window but all close to the lower/upper bands? How (3) and (4) can be detected error-free?

(5) I checked the author's prior paper that the ACAM design was proposed there. I am curious about the robustness of the proposed ACAM against process/device-to-device variations. It was not discussed in that paper. Getting four or eight states (2/3 bits) from a single device is prone to be affected by variations and results in erroneous write/read operations. Does this concern make sense? Please comments on it.

(MA-3) The paper story should be improved.

(MI-1) Fig. 3.d is not clear and might not be necessary to be in the paper.

(MI-2) In Fig.4.a, what does the sense Amps (there is a typo in the figure! "Semse") look like? The authors should depict the circuit diagram (maybe in the supplementary?).

Reviewer #3:

Remarks to the Author:

This paper demonstrates a RRAM-CAM based in-memory hardware accelerating tree-based

machine learning models with a non-volatile analog CAM hardware previously proposed. Overall, this research is fairly interesting and provides quite impressive improvement results. Still, this reviewer thinks several argument points remain in evaluations and important details.

1. The analog CAM with memristor based in-memory computing architecture can be considered as an original contribution of this paper. However, the proposed architectural optimization which is reordering the feature vector in CAM array, is a simple idea. So, the reviewer thinks the novelty of this work is marginal and it can be considered as an incremental work utilizing the previous CAM cell circuit [1]. The author should emphasize/highlight the original contribution of this work.

2. In figure 5, the author has decided the tile size of 16X480 considering power consumption and throughput (not the network size) of the analog CAM system. The evaluation results in figure 5 show that SL static power consumption seems to be dominant. However, for the comparison of the power consumption of the entire system, "Energy per decision" values also need to be compared since the DT (decision tree) node are related with array size. In addition, providing the power breakdown (power consumption of different components) of the entire system makes this issue more clear.

3. Even though analog CAM can perform the tree based machine learning operations inside memory, the area of the analog CAM cell used in this work is larger than that of SRAM cell, which incurs large area overhead. Therefore, it would be better to add area analysis of the overall architecture including analog CAM.

4. The detailed neural network structure has been not presented in this paper. Considering that hardware architecture such as memory size and data mapping are determined by neural network structure, , it is necessary to show the detailed neural network.

5. In table 1, the proposed ML accelerator is designed using 16nm CMOS technology while the previous works used 40nm~65nm CMOS technology. Those accelerators should be compared under same basis.

We thank the editor and reviewers for their careful review of our work. Based on these comments and suggestions we have revised the manuscript. All changes to the text are highlighted in the revision and we have provided point by point responses below. The main changes made to the manuscript are:

- a) We included discussion on the trillions of operations per seconds (TOPS) and the inherited calculation of energy efficiency (TOPS/W) and area efficiency (TOPS/mm²) compared with existing technologies.
- b) We changed the reference technology of the benchmark from 16 nm to 65 nm in accordance to the other ASIC for comparison in Tabel I
- c) We normalized the benchmark to the node operation (i.e. threshold of decision tree) to compare different algorithmic realization more fairly
- d) We included additional details on the circuit and ML algorithms
- e) We improved the main text and reference

Reviewer 1

The paper proposes the design of accelerators for Random Forest (RF) algorithms using their previously proposed (Ref. 36) analog memristor-based content addressable memory (CAM). Large RF inference models are mapped to arrays of analog CAM. Measured data from Ref. 36 is used to calibrate the CAM cell model in 180nm CMOS. Then, using circuit models of the 180nm CAM cell as well as extrapolated models in 16nm CMOS, the authors demonstrate significant gains in throughput (3-orders-of-magnitude) and energy-efficiency (2-orders-of-magnitude) gains over previously proposed works. These results are shown in simulations.

It is **refreshing to see a focus on something other than deep neural networks (DNNs). RF are powerful ML models but haven't been studied from the perspective of implementation.** Furthermore, **such models have the advantage of robustness and interpretability** as is mentioned by the authors.

We thank the reviewer for the positive comments on our work, indeed we consider the novelty of accelerating tree-based and interpretable ML algorithms as a main breakthrough for in-memory accelerators. Below, please see a point by point response to each comment.

1. On a related note, the manuscript mentions RF and DT as two distinct ML models when in fact RF is an ensemble of DTs. It would be better to drop the mention of DT unless it is used as a RF component.

We thank the reviewer for the correction. **We have modified the text accordingly** by inserting the following sentence in the introduction

“...tree-based methods, such as decision trees (DT) and their ensembles, for example random forest (RF) methods...”

2. A major issue with this paper is the reliance on simulations using circuit models to prove the benefits of the proposed architecture. While there is nothing wrong with the approach, in my opinion, this type of work is better suited for other journals in the devices area for example IEEE JxCDC. Nature papers need to represent a major breakthrough of sorts involving basic sciences. It was difficult to find any such breakthrough in this paper even though the material itself was interesting.

We thank the reviewer for the comment. The goal of this work is not to demonstrate an integrated circuit performing DT/RF inference, which would likely be appropriate for another venue. Instead, the core **breakthrough** of this work is a differentiated in-memory computing concept, an important application area, and a validation of substantial performance advantage based on grounded analysis. The differentiated concept here is the efficient mapping of tree models to analog CAM circuits, with root to leaf paths naturally mapped to CAM rows. We believe that this offers a substantial breakthrough and contrast, as most accelerator research focuses on Deep Learning. However as pointed out by the reviewer, “RF are powerful ML models but haven't been studied from the perspective of implementation. Furthermore, such models have the advantage of robustness and interpretability...” which are main requirements for several critical applications. We also believe that this idea can open new research directions similar in impact to the first demonstrations¹ of matrix vector multiplications performed in resistive crosspoint arrays to speed-up Deep Learning. This in-memory computing approach and application area has largely come to dominate research by this community. Such an approach offers no speed-up to the important class of tree-based models here, which are not dominated by matrix operations. Instead, highly irregular memory look-up patterns

are addressed with the analog CAM approach described here thanks to the capability to store ranges of values, and a don't care 'X' entry which enables compression and efficient representation. Thus, we show a new computing concept on a new hardware primitive (crosspoint arrays were invented decades ago, while the enabling concept of the analog CAM is very recent). We further quantified the expected benefits compared to conventional and custom accelerators. The analysis is not based on simple spice-like circuit simulations but on a reliable custom behavioural model which represents the non-idealities of actual taped out analog CAM circuits² in 180nm technology with memristors integrated in a back-end-of-the-line (BEOL) process in our lab. These circuits were measured, and their operation validated and understood. We then developed a compact model tailored to the parasitics extracted from the chip hardware, and validated against measured chip operation, in order to model the resistance and capacitance of each line. The model is fitted with data experimentally measured on the taped-out chip hardware, as is shown in Figure 2. With this approach, we take into account the device and circuit non-idealities which feed into our larger-scale architecture simulations, comprising a total of 194000 analog CAM cells. In these large-scale simulations, **we also utilize the experimentally measured statistics of conductance values of the memristors across 4096 devices**, and an example of this experimental data is shown in the cumulative density function distributions shown in Fig. 1f. We thus performed ex-situ computing with actual memristor conductance read from a taped-out chip. While we agree that a fully integrated chip (analog CAM circuits plus all peripherals) would be ideal, this is beyond the scope of this present work and journal. Instead, we emphasize our approach incorporates experimentally measured data and circuit characteristics to produce a model allowing the scaling to larger, more relevant real-world problems.

3. Related to Comment 2, the comparisons in Table 1 are highly unfair. For one, this table compares ASIC implementations in Refs. 13, 15, and 26 with simulations results using circuit models of the CAM cell. A well-known truism is that real-life implementations don't measure up to simulations in terms of the results. The work would be impactful if the authors were to build a complete CAM array including all peripherals and then report results. In that case, even a 4X-10X gains in energy efficiency would be very impressive provided these are done at iso-accuracy.

We thank the reviewer for this comment. As pointed out in the previous comment, we apologize for the confusion - our work does not intend to present a custom integrated chip, but presents a new in-memory acceleration concept and supporting performance analysis. As pointed out in the previous comment, analog CAM models are based on reliable hardware measurements while all the peripherals (pre charge circuit, sense amplifier, digital to analog converters...) are simulated in Virtuoso environment and included in the simulations that yielded Table 1. Circuit design, full VLSI tape-outs, and memristor integration are future substantial works on their own. However, given the current importance in both explainable AI such as in tree-based models³⁻⁸ and excitement for computing with CAMs^{2,9,10} we believe the present work is of substantial and timely impact even without an ASIC tape-out. This is also a common and established practice in related works by the in-memory computing community¹¹⁻¹⁷

Finally, we note that the acceleration of throughput by three orders of magnitude gives significant margin for error while still yielding compelling performance advantage. This is due to the strong benefit of having a non-volatile analog CAM cell for accelerating the root to leaf paths rather than the costly non-uniform memory access patterns.

4. Related to Comment 3, comparison with Ref. 15 is very unfair since it is a digital processor for video object recognition at 60 fps. Data volumes needed to execute such computations is

much larger than classification task on the KUL Belgium data set using RF as done in the paper. Similarly, fully programmable processors such as Intel (Ref. 45) or Tesla (Ref. 45) or Virtex (ref. 45) is very easy to beat in performance and power. Comparison with Ref. 26 would have been reasonable but, as mentioned in Comment 3, this is a real-life IC with measured results. Energy and throughput need to be compared for the same task (classification) and using the same inference model and at the same accuracy.

We thank the reviewer for the comment, and have improved our comparisons given these suggestions. We normalized the performance metric to the cost for inferencing a single node, such that the comparison is fairer, an approach inspired by the reference¹⁸. In particular, for the KUL traffic sign dataset during inference we need to transverse ~ 4000 nodes, corresponding to each threshold of each DT in a forest. We also agree with the reviewer that it is easy to beat in performance and power a fully programmable processor. However, we emphasize our in-memory computing approach outperforms not only mainstream processors but also application specific hardware acceleration in ML model inference¹⁸. We agree with the reviewer that it is important to compare for the same task. In fact, by normalizing the workloads at the node traversing operation, we can directly compare with other ASIC implementations.

We revised the text accordingly in the performance evaluation section and also to respond to another reviewer comment, we have provided the benchmark at the same technology node (i.e. 65 nm) of the reference¹⁸ for a more fair comparison.

5. The paper is missing comparison using arithmetic efficiency metrics such as TOPS (tera operations per second) and TOPS/W. Another important metric is computational density in TOPS/mm². This is a specially important metric since one of the purported advantages of RRAMs is their density

We thank the reviewer for the comment. TOPS are typically used for benchmarking at the architecture level, rather than at system¹⁹. However, we agree with the reviewer that adding performance based on TOPS will improve the clarity of our results for a broader range of readers.

As it is possible to see from the reference¹⁸ in the case of DT and RF inference, the basic operation to consider is the number of nodes (thresholds) that an inference needs to traverse (compare) before reaching the final classification answer. Based on the reviewer suggestion we thus normalized our metrics to the number of nodes we traverse. For the RF evaluation, we used 15 trees and a maximum depth of 10, which corresponds to $N_0 = 4000$ nodes that need be evaluated. Considering that analog CAM can map 4 bits for each cell, we consider a 4 bits operation the minimum operation size. From that we can compute in the pipelined architecture whose shows a throughput $f = 333 \frac{MDec}{s}$, the performance metric τ

$$\tau = N_0 \cdot f = 4000 \cdot 333 \frac{MDec}{s} = 1330 \text{ TOPS}$$

Similarly, the energy efficiency η can be computed as

$$\eta = \frac{\tau}{Power} = \frac{1.33 \text{ TOPS}}{427 \text{ mW}} = 3.11 \frac{\text{TOPS}}{W}$$

These numbers should be compared with the best result available¹⁸ which reports 2.89 GOPS and 407 GOPS/W performing the inference task on the same dataset we considered and with similar accuracy. Note that this calculation considers the maximum hardware efficiency (optimum mapping, full memory usage) as typically done for benchmarking AI inference systems. This is different from

the algorithmic efficiency of the RF implementation. In fact, given a balanced tree with ν nodes, the average number of operation that a conventional (or, non-parallelized) processor needs to perform is $h = \log_2 \nu$ to reach a leaf, which corresponds to the height of the tree. By considering this, algorithmic performance can be computed. The algorithmic equivalent number of operations N_1 analog CAM performs at each inference cycle is

$$N_1 = N_{trees} \cdot \log_2 \nu = 15 \cdot \log_2 256 = 120$$

While the reference¹⁸ would perform $N'_1 = 317$. All figures of merit can be computed then from an algorithmic perspective as reported for simplicity in the table below.

Accelerator	Type of metric	Number of nodes	τ	η
aCAM	Hardware	4000	1.33 TOPS	3.11 TOPS/W
SRAM	Hardware	1984	2.89 GOPS	407 GOPS/W
aCAM	Algorithmic	120	33.96 GOPS	80.28 GOPS/W
SRAM	Algorithmic	317	231 MOPS	32.53 GOPS/W

This calculation highlights that the main benefit of our implementation is strongly due to the massive parallel operation.

Given that we don't have an optimized cell layout at the same technology node to compute the area efficiency, we compute it at 16 nm node with a previously shown 6T2M analog CAM layout² which is reported in Fig. R1 with a resulting area of $A_{6T2M} = 0.51 \mu m^2$

Figure R1 – Layout of 6T2M analog CAM cell in 16 nm technology²

To perform RF inference, the state of each match line (ML) needs to be stored in a register. The register area per bit at 16 nm is $A_{reg} = 5.5 \mu m^2$. After collecting all the ML results, the outputs need to be aggregated with an AND operation before performing the majority voting. Every logic per bit has an area of $A_{logic} = 0.28 \mu m^2 \cdot 2 = 0.56 \mu m^2$ where the factor 2 is to consider the logic placement utilization due to routing overhead. We previously² estimated the area of a current steering digital to analog converter (Supplementary Information Figure 7) to be $A_{DAC} = 10 \cdot A_u \cdot N_{ch} + 5 \cdot A_{mir}$ where $A_u = 0.046 \mu m^2$ is the area of a transistor with a single finger, $A_{mir} = 0.061 \mu m^2$ is the area of the current mirror transistors and N_{ch} the total number of desired channels

Hence, the total area occupation of our accelerator in 16 nm technology can be computed as:

$$A_{aCAM} = [N_{arrays} \cdot (H \cdot W \cdot A_{6T2M} + H \cdot A_{reg} + H \cdot A_{AND} + H \cdot A_{MAJ}) + A_{DAC}] \cdot (1 + OH) = 0.266 \text{ mm}^2$$

Where $N_{arrays} = 29$ is the total number of arrays in the architecture, $H = 480$ and $W = 16$ is the number of rows and columns in each array respectively, $N_{ch} = W \cdot N_{arrays}$ for the full pipelined implementation, A_{AND} is the area of an AND gate, $A_{MAJ} = 3A_{AND}$ the area of a majority voting gate, and $OH = 0.2$ is an extra overhead factor for placement and routing. With a fixed clock frequency of $f_{clk} = 1 \text{ GHz}$ the throughput at 16 nm remains the same of the previously calculated at 65 nm, leading to the same performance τ . Thus the area efficiency α can be computed as

$$\alpha = \frac{\tau}{A_{aCAM}} = \frac{1.33 \text{ TOPS}}{0.266 \text{ mm}^2} = 5 \frac{\text{TOPS}}{\text{mm}^2}$$

To compare this result with the reference¹⁸ which is based on 65 nm technology, we need to scale it first to 16 nm. In the TSMC process high density 6T SRAM bit cell has an area of $0.499 \mu\text{m}^2$ at 65 nm and $0.07344 \mu\text{m}^2$ at 16 nm. Hence the scaling factor is $\frac{0.499}{0.07344} \sim 6.8$. Note however that in the reference¹⁸, the bit cell area size is $2.11\mu\text{m} \cdot 0.92\mu\text{m} = 1.94 \mu\text{m}^2$ which is several times larger than the high density bit cell. This is due to upsizing of the pass-gate and pull-down transistors to reduce the resistance of the bit line discharge path, which helps to increase the bit line voltage discharge swing. It can be expected that the 16 nm bit cell would need to be upsized in a similar way, keeping the ratio of ~ 6.8 . Thus we can estimate the area of the 6T SRAM bit cell at 16 nm as $A_{6T} = \frac{1.94\mu\text{m}^2}{6.8} = 0.285 \mu\text{m}^2$. Similarly, we can compute the scaling of the logic. Approximate metric for density scaling is the product of contacted poly pitch (CPP) and minimum metal pitch (MXP). In TSMC process $CPP \cdot MXP = 0.16 \cdot 0.18 \mu\text{m}^2$ at 65 nm and $CPP \cdot MXP = 0.09 \cdot 0.064 \mu\text{m}^2$ at 16 nm, leading to a scaling factor $\frac{0.16 \cdot 0.18}{0.09 \cdot 0.064} = 5$. Nevertheless, improvements in cell library design for advanced technology nodes can further improve this scaling ratio. For example, TSMC reports 2.35x density improvement from 65 to 40 nm, 2x improvements from 40 to 22 nm, and 1.44x improvements from 22nm to 16 nm, yielding to a cumulative logic density improvement of 6.8x from 65 to 16 nm. For considering a best-case scenario for designing the SRAM logic at 16 nm we consider an logic density scaling factor of 6.8x. Figure R2 shows the layout at 65 nm of the SRAM based accelerator in the reference¹⁸.

Figure R2 – Layout of SRAM based accelerator¹⁸ with highlighted area for the comparison.

We can directly measure the memory area (bit cell array) and logic (CTRL), which at 65 nm results in a total of $A_{SRAM65} = 0.56 \text{ mm}^2$. By applying the aforementioned scaling rules we can compute the area at 16 nm as $A_{SRAM16} = \frac{A_{SRAM65}}{6.8} = 0.082 \text{ mm}^2$ which leads to an area efficiency of

$$\alpha_{SRAM} = \frac{2.89 \text{ GOPS}}{0.082 \text{ mm}^2} = 35.24 \frac{\text{GOPS}}{\text{mm}^2}$$

While the area occupation for SRAM based accelerator is significantly lower compared to analog CAM accelerator, the resulting area efficiency is also drastically lower. This is due to the full parallelism that it is possible to achieve with the analog CAM.

We conclude that the analog CAM accelerator is 460x more performant, 7.6x more energy efficient and 142x more area efficient than the SRAM based accelerator considered as reference¹⁸.

We added this discussion in the supplementary materials.

- The proposed CAM cell is a 6T-2M cell with two resistive devices and 6 NMOS transistors. Transistors T1 and T3 along with their memristive loads act as threshold gates that turn ON at different values of DL. The rest of the functionality is implemented using NMOS transistors. This role of the memristive device seems not as central as it is in an architecture such as a cross-point. Both the area and power seem to be dominated by the CMOS switches. If Figure 2a were to mark the RRAM devices, this issue would be clear. Is this a good use of memristive devices? One benefit of NVM devices is the increase in computational density. The proposed CAM cell seems to miss out on this opportunity.

We thank the reviewer for the comment. We apologize for being unclear, we would like to first clarify a few concept. Analog CAM cell is not made of 6 NMOS transistors + 2 memristors. It is rather a 4 NMOS + 1 CMOS + 2 memristor design, where the CMOS is a conventional inverter gate. Secondly consider that also for crosspoint arrays a transistor has to be inserted in series with each memristor to allow the addressability and control the compliance current required in the analog set operation. Thus also in conventional crosspoint memory the transistor area is dominant . Finally, power consumption is not dominated by the CMOS circuitry, but the main contribution is due to the static power consumption in the voltage divider between the memristor and the series memristor. The power consumption would in fact benefit a lot with improved memristor conductance range (i.e. lower conductance).

Memristive devices are central in the analog CAM operation. We partially answered this issue in the previous comment. However, it is useful to compare this result with the ternary CAM (TCAM) that we previously showed⁹. Continuously tuning the conductance allows for efficient representation of the real-valued thresholds in decision trees in a compact fashion directly in hardware. The range capability also enabled by the analog CAM circuit is core to this mapping and representation, to map a range similar to conventional ternary CAM (TCAM) requires several rows as already shown². Figure R3 reports an example taken from ref 35 of the dimension of the memory for mapping a search operation with TCAM and analog CAM with different memristor bit precision, demonstrating a reduced area.

Figure R3 CAM tables for searching a range between 385 and 58630, with (a) TCAM, (b) 3-bit analog CAM, (c) 4-bit analog CAM and (d) 8-bit analog CAM².

It is not just the density, but it is a functional difference that the analog tunability of memristors + the analog CAM concept allow.

- The manuscript is well-organized overall. However, it has a number of awkwardly put sentences, e.g., second sentence on page 1: “However, DNN impact is limited for a range of applications where...” when the authors mean to say that “DNNs are unsuitable for applications where explainability is critical”. Similar sentences are interspersed throughout. The authors are urged to revise such statements.

We thank the reviewer for the comment. **We revised the text accordingly.**

- Typos – page 1, second para, line 8: “...is used in analog...” needs to be “...are used in analog...” since the reference is being made to “(NVM) devices...”.

We thank the reviewer for the comment. **We revised the text accordingly.**

Reviewer 2

This paper proposes to use NV ACAMs to implement DT/RF algorithms in hardware. Here are my major (MA)/minor (MI) comments:

We thank the reviewer for carefully reading the manuscript and provide comments. Below is a point by point reply to the comments.

1. (MA-1) How much are the DT/RF algorithms still useful and practical?

We may find more recent/efficient algorithms. The references that the authors have cited do not convince me. Citing Ref.2 and Ref.3 surprises me. Ref. 4 is a survey paper and has cited two (2016/2017) papers for DT algorithms. I could not access ref.5. Ref.s 6-8 are pretty old, and the only prestigious/recent reference about the importance of DT/RF algorithms is ref.9.

We thank the reviewer for the comment. We apologize for not being clear enough in explaining the need for DT/RF accelerators. While a lot of research is currently focused on deep neural networks, tree-based algorithms are the preferred methods for broad range of government³⁻⁵ and industry related applications (ref 3), where inspectability and explainability are as important as the accuracy of the model itself. In citing ref. 3, we demonstrate in fact that these tree-based models are in use broadly as state-of-the-art approaches by machine learning practitioners today. They are the preferred approach across a wide range of disciplines, highlighting their practical use. We agree that some works cited were older and less relevant here, and have thus removed them. Instead we cited recent relevant works where tree-based algorithm are used in clinical tasks^{7,8}.

We updated the text accordingly.

2. (MA-2) How much is an ACAM array flexible for various DTs? Each tree can have different shapes and different numbers of nodes.

We thank the reviewer for their comments – in fact a strength of the analog CAM approach is that it handles these tree differences flexibly and directly within the hardware! By inserting “don’t care” nodes in root-to-leaf paths for missing features, the ACAM allows for mapping an arbitrary root to leaf path of a DT. However, in this zeroth level mapping, the number of features that can be looked up (inferred) at each time step is limited by the maximum number of columns that an ACAM array can have (in our case ~50 columns due to match line parasitic capacitance limitation). To overcome this challenge, we developed a tiled architecture (illustrated in Fig. 4e) where if the feature vector is larger than the number of columns it is divided in multiple array whose output is joined with an AND operation in postprocessing. Note that also other techniques can be used, with tiled rows are turned on only if the path traversal before them is active. These architectural approaches for improving performance are directly enabled by the in-memory mapping of the DT and RF models to the analog CAM hardware.

We updated the text accordingly.

3. (MA-3) I am not sure about the practicality of an ACAM array to realize a DT algorithm.
 - a. Data input can be a voltage value between 0 to 1V or 0 to 1000mv. 0-1V is very limited. 0-1000mv might not be that practical to cover any value range for a DT. e.g., what if we need to check $523\text{mV} < \text{data} < 789\text{mV}$? How can we program the memristors to give us that accurate range? If a degree of inaccuracies is acceptable by the algorithms, it should be discussed/evaluated in the paper.

- b. How to store an 'X' state to an MCAM cell? I think (not sure) the lower band cannot be as low as 0V.

We thank the reviewer for the comment and agree that memristor conductance programmability and accuracy is a critical factor to consider in this approach. While not highlighted explicitly within the text, we are very conscious of these considerations and thus utilize experimentally measured conductance distributions in our evaluations. Our memristor technology has a programmability of 4 bits in each cell as we showed in the distribution of Fig. 1f. This measurement is conservative, in fact we considered only conductance states that showed perfectly linear conduction behaviour (see Fig. 1g) but our memristors can be programmed in a wide range on states with a relatively low noise figure, i.e. conductance variation over time²⁰. (Figure R4,R5)

Figure R4 Demonstration of large window and analog tunability of the integrated memristor devices²⁰.

Figure R5 Amount of noise at different programmed analog states²⁰.

We note as well that the precision can be increased by using multiple cells for representing a single range. For the task we consider (RF inference), however, the precision is not a strict requirement thanks to the collective/ensemble behaviour of this type of ML model. We see this relaxed precision requirement experimentally in Fig. 4c, which shows the (surprisingly small) accuracy loss compared to floating point precision as function of the number of bits used. Thus, we considered a 4 bits memory cell and 4 bit input DAC.

As pointed out by the reviewer, it is not possible to map the information in a range starting from 0 since it would not be possible to program a don't care (X) on the left bound. Even if our memristor allows very low conductance for a practical implementation we considered a don't care on the left bound equal to 0.1 μ S and a don't care on the right bound equal to 250 μ S which allow us to safely store data whose input is in the range 0.1V to 1.1 V, which corresponds to a DAC least significant bit LSB = 62.5 mV. While this precision can be easily reached in hardware, memristors still suffer from

variation and CMOS circuitry can induce analog noise. Thus, we studied the accuracy as function of the conductance (thus threshold) variation, which a good indicator of the overall noise resilience. As shown in Fig. 4b, the accuracy does not change significantly for a standard deviation of noise up to 10%. We conclude the system is robust against noise.

- c. Having a rather large row, let's say a 64-cell row, how much the array can detect the worst-case mismatch cases? e.g., we have 63 matched cells, and we do have one mismatch cell while the input query is very close to the lower/upper band. Can the row detect this marginal case as a mismatch case?
- d. What about the worst-case match cases? We have 64 cells, which all are inside the matching window but all close to the lower/upper bands? How (3) and (4) can be detected error-free?

We thank the reviewer for the comment. First of all we would like to point out that our current design does not allow for a number of rows larger than 50, without degrading the latency performance in detecting one bit mismatch. In our previous work² we characterized the analog CAM array level latency for 1-bit mismatch as a function of the number of columns W as shown in Figure R6.

Figure R6 Latency as a function of the number of columns for 16 nm analog CAM².

The results show that the latency increases sub linearly with W , and we considered as a maximum latency 100 ps corresponding to $W = 50$ at 16 nm technology node.

Nevertheless, in this work we used range encoding to map DT root to leaf path into analog CAM rows, which is different from a multibit programming. In fact, ranges comprising >1 -bit or accepting all values below (or above) a given one need to be programmed. For this reason, to assess the reviewer question we characterized the analog CAM range encoding behaviour, by considering its immunity to noise. Note also that the marginal cases are of lower impact for ensemble tree models due to the voting behaviour. To show the noise immunity, we programmed 50 columns conductance G_{left} and G_{right} with random target ranges X_{left} and X_{right} including a random population of don't cares with 10% probability of appearance. Then we evaluated the output voltage after sensing the ML as function of the % mismatch of the input value compared with the stored threshold.

Figure R7 (a) Conductance programmed in 50 analog CAM columns. (b) Sensed voltage as function of [%] mismatch compared with the stored threshold

Figure R7 shows the result demonstrating that our system allows variation of of the input data up 12%. This is mostly due to the fact that the left boundaries are smooth compared to the right boundaries where an inverter is inserted between the voltage divider and the output transistor gate. In our application, if a single feature differs more than $\sim 12\%$ from the threshold stored, the corresponding row is surely a mismatch. By analogy it can be calculated the total number of levels that can safely be stored in an analog CAM is $1/0.12 = 8.3$ which matches with our initial assumption of 3 bit of capacity and compatible with our memristor technology. To increase the number of level (thus sensitivity to 1 bit mismatch) the analog CAM should be re-designed to have steeper boundary on the left side, for example by inserting a chain of two inverters between the voltage divider and the output transistor.

- e. I checked the author's prior paper that the ACAM design was proposed there. I am curious about the robustness of the proposed ACAM against process/device-to-device variations. It was not discussed in that paper. Getting four or eight states (2/3 bits) from a single device is prone to be affected by variations and results in erroneous write/read operations. Does this concern make sense? Please comments on it.

We thank the reviewer for the comment. Indeed process and device to device variation are a main concern when dealing with emerging memory devices. We measured 4 bits on 64x64 1T1R integrated memristor crossbar array which allowed us to consider both CMOS and memristor process variation as shown in Figure 1f. Data extracted from this distribution were used for simulating the inference process in the RF task for having an accurate error evaluation considering all CMOS and memristive variations.

We updated the text to clarify this point.

4. (MA-3) The paper story should be improved.

We thank the reviewer for the comment. In this work, we presented the core breakthrough of our differentiated in-memory computing concept, which enables the efficient mapping of tree models to analog CAM circuits, with root to leaf paths naturally mapped to CAM rows.

First we showed an analog CAM compact model for describing the post tape-out cell circuit behaviour including parasitics and non idealities. Then, we developed an architecture for handling large scale models such as random forests (RF). Finally, we optimized the sizing of analog CAM arrays for a particular tasks (KUL Belgium Traffic Signs dataset classification) and compared the result with state of the art ASIC implementations.

We believe that the importance of the ML models targeted in this paper, the novel in-memory computing approach described to speed-up this problem class, and the quantitative performance analysis grounded in an actual chip tape-out offer a compelling story. We welcome comments and questions to improve it further (as the reviewers have already inspired). We would be happy to address more detailed requests or suggestions from the reviewer to improve the paper story further.

5. (MI-1) Fig. 3.d is not clear and might not be necessary to be in the paper.

We thank the reviewer for the comment. We apologize for the unclear figure. Figure 3d shows what happens inside each cell of the analog CAM array during inference of a decision tree. On the left (blue lines) it is possible to see the current flowing in each cell of a 7x4 analog CAM array as function of input voltage on the DL corresponding to the feature vector, and on the right the voltage on the ML as function of time for a particular feature vector (indicated by the vertical dashed lines). With this plot, it is possible to characterize the analog CAM array response to an arbitrary input. We believe it also helps to demonstrate the cell-level behavior and collective behavior of the analog CAM array.

6. (M1-2) In Fig.4.a, what does the sense Amps (there is a typo in the figure! "Semse") look like? The authors should depict the circuit diagram (maybe in the supplementary?).

We thank the reviewer for the comment.

We added a schematic diagram of the sense amplifier in supplementary information and we corrected the typo in the figure.

Reviewer 3

This paper demonstrates a RRAM-CAM based in-memory hardware accelerating tree-based machine learning models with a non-volatile analog CAM hardware previously proposed. Overall, this research is **fairly interesting** and provides quite **impressive improvement results**. Still, this reviewer thinks several argument points remain in evaluations and important details.

We thank the reviewer for the positive comments about our work and for the time in carefully reading and reviewing the manuscript. Please find below a point to point reply to the comments and issues raised.

1. The analog CAM with memristor based in-memory computing architecture can be considered as an original contribution of this paper. However, the proposed architectural optimization which is reordering the feature vector in CAM array, is a simple idea. So, the reviewer thinks the novelty of this work is marginal and it can be considered as an incremental work utilizing the previous CAM cell circuit [1]. The author should emphasize/highlight the original contribution of this work.

We thank the reviewer for the comment. We would like to highlight the original contribution of our work here. The core breakthrough of this work is a differentiated in-memory computing concept for the efficient mapping of tree models to analog CAM circuits, with root to leaf paths naturally mapped to CAM rows. We believe that this offers a substantial breakthrough and contrast, as most accelerator research focuses on Deep Learning. However tree-based models are powerful ML models but haven't been studied from the perspective of implementation for in-memory computing. Furthermore, such models have the advantage of robustness and interpretability, which are main requirements for several critical applications. We also believe that this idea can open new research directions similar in impact to the first demonstrations¹ of matrix vector multiplications performed in resistive crosspoint arrays to speed-up Deep Learning. This in-memory computing approach and application area has largely come to dominate research by this community. Such an approach offers no speed-up to the important class of tree-based models here, which are not dominated by matrix operations. Instead, highly irregular memory look-up patterns are addressed with the analog CAM approach described here thanks to the capability to store ranges of values, and a don't care 'X' entry which enables compression and efficient representation. Thus, we show a new computing concept on a new hardware primitive (crosspoint arrays were invented decades ago, while the enabling concept of the analog CAM is very recent). Further, given the current importance in both explainable AI such as in tree-based models³⁻⁸ and excitement for computing with CAMs^{2,9,10} we believe the present work is of substantial and timely impact.

While the concept is simple (similar to the above matrix crosspoint array concept), the result is extremely compelling from a performance standpoint outperforming the state-of-the-art by orders of magnitude (see main table of manuscript). And this performance comes from the unique features of the analog CAM such as **range storing, analog search**. Further, while the massive compression from the 'X' or 'don't care' capability may seem simple – and the architectural optimization with the reordering – it is one of the main features of the analog CAM hardware and our work here is the first demonstration of a **hardware aware compression algorithm** for mapping trees on CAM hardware. We don't believe these techniques are obvious, and we also highlight this is the first time this novel in-memory computing approach has been described to accelerate inference for this important problem class of tree-based models. And in addition to the idea, the quantitative performance analysis (grounded in an actual chip tape-out) supports the work with compelling results compared to the state-of-the-art.

We improved the main text by highlighting these concepts.

2. In figure 5, the author has decided the tile size of 16X480 considering power consumption and throughput (not the network size) of the analog CAM system. The evaluation results in figure 5 show that SL static power consumption seems to be dominant. However, for the comparison of the power consumption of the entire system, "Energy per decision" values also need to be compared since the DT (decision tree) node are related with array size. In addition, providing the power breakdown (power consumption of different components) of the entire system makes this issue more clear.

We thank the reviewer for the comment. We agree with the reviewer that energy per decision would be the metric for tile sizing. However, here our goal is to show that an in-memory implementation with analog CAM can overcome the performance of the state of the art. For a fair comparison, we evaluated all figures of merit normalized to the number of nodes, as it was previously done¹⁸. We added a figure in the supplementary material with the energy per decision.

Regarding the power breakdown, we added a pie-chart comparison of the power consumption of each component in 180nm and 65nm technology in the supplementary materials and changed the main text accordingly.

3. Even though analog CAM can perform the tree based machine learning operations inside memory, the area of the analog CAM cell used in this work is larger than that of SRAM cell, which incurs large area overhead. Therefore, it would be better to add area analysis of the overall architecture including analog CAM.

We thank the reviewer for the comment. Our 6T2M analog CAM cell area is indeed larger than the usual 6T SRAM area which was used for storing thresholds in the reference¹⁸. To measure we compared a 16 nm layout area we previously showed² (Figure R1 above) $A_{6T2M} = 0.51 \mu m^2$ with the size of SRAM cell in 16 nm technology. In TSMC process high density 6T SRAM bit cell has an area of $0.499 \mu m^2$ at 65 nm and $0.07344 \mu m^2$ at 16 nm. Hence the scaling factor is $\frac{0.499}{0.07344} \sim 6.8$. Note however that in the reference¹⁸, the bit cell area size is $2.11 \mu m \cdot 0.92 \mu m = 1.94 \mu m^2$ which is several times larger than the high density bit cell. This is due to upsizing of the pass-gate and pull-down transistors to reduce the resistance of the bit line discharge path, which helps to increase the bit line voltage discharge swing. It can be expected that the 16 nm bit cell would need to be upsized in a similar way, keeping the ratio of ~ 6.8 . Thus we can estimate the area of the 6T SRAM bit cell at 16 nm as $A_{6T} = \frac{1.94 \mu m^2}{6.8} = 0.285 \mu m^2$, which is indeed smaller than A_{6T2M} . However, analog CAM can not only store multiple bits (i.e. 4) but it allows to store ranges. Moreover, thanks to the non volatility the throughput can be speeded up notably. A fair area comparison between SRAM and analog CAM accelerator should be done based on the operation efficiency. To do that we first compute the total area of the RF accelerator with analog CAM. To perform RF inference, the state of each match line (ML) needs to be store in a register. The register area per bit at 16 nm is $A_{reg} = 5.5 \mu m^2$ After collecting all the ML results, the outputs need to be aggregated with a AND operation before performing the majority voting. Every logic per bit has an area of $A_{logic} = 0.28 \mu m^2 \cdot 2 = 0.56 \mu m^2$ where the factor 2 is to consider the logic placement utilization due to routing overhead. We previously² estimated the area the current steering digital to analog converter (Supplementary Information Figure 7) to be $A_{DAC} = 10 \cdot A_u \cdot N_{ch} + 5 \cdot A_{mir}$ where $A_u = 0.046 \mu m^2$ is the area of a transistor with a single finger, $A_{mir} = 0.061 \mu m^2$ is the area of the current mirror transistors and N_{ch} the total number of desired channels

Hence, the total area occupation of our accelerator in 16 nm technology can be computed as:

$$A_{aCAM} = [N_{arrays} \cdot (H \cdot W \cdot A_{6T2M} + H \cdot A_{reg} + H \cdot A_{logic}) + A_{DAC}] \cdot (1 + OH) = 0.266 \text{ mm}^2$$

Where $N_{arrays} = 29$ is the total number of arrays in the architecture, $H = 480$ and $W = 16$ is the number of rows and columns in each array respectively, $N_{ch} = W \cdot N_{arrays}$ for the full pipelined implementation and $OH = 0.2$ is an extra overhead factor for placement and routing.

Then, the number of operation per seconds should be calculated. As it possible to see from the reference¹⁸, in the case of DT and RF inference, the basic operation to consider is the number of nodes (thresholds) that an inference needs to traverse (compare) before reaching the final classification answer. Based of another reviewer's suggestion, we thus normalized our metrics to the number of nodes we traverse. For the RF evaluation, we used 15 trees and a maximum depth of 10, which corresponds to $N_o = 4000$ nodes that need be evaluated. Considering that analog CAM can map 4 bits for each cell, we consider a 4 bits operation the minimum operation size. From that we can compute in the pipelined architecture whose shows a throughput $f = 333 \frac{MDec}{s}$, the performance metric τ

$$\tau = N_o \cdot f = 4000 \cdot 333 \frac{MDec}{s} = 1.33 \text{ TOPS}$$

Thus the area efficiency α can be computed as

$$\alpha = \frac{\tau}{A_{aCAM}} = \frac{1.33 \text{ TOPS}}{0.266 \text{ mm}^2} = 5 \frac{\text{TOPS}}{\text{mm}^2}$$

To compare this result with the reference¹⁸ which is based on 65 nm technology, we need to scale it first to 16 nm. In TSMC process high density 6T SRAM bit cell has an area of $0.499 \mu\text{m}^2$ at 65 nm and $0.07344 \mu\text{m}^2$ at 16 nm. Hence the scaling factor is $\frac{0.499}{0.07344} \sim 6.8$. Note however that in the reference¹⁸, the bit cell area size is $2.11 \mu\text{m} \cdot 0.92 \mu\text{m} = 1.94 \mu\text{m}^2$ which is several times larger than the high density bit cell. This is due to upsizing of the pass-gate and pull-down transistors to reduce the resistance of the bit line discharge path, which helps to increase the bit line voltage discharge swing. It can be expected that the 16 nm bit cell would need to be upsized in a similar way, keeping the ratio of ~ 6.8 . Thus we can estimate the area of the 6T SRAM bit cell at 16 nm as $A_{6T} = \frac{1.94 \mu\text{m}^2}{6.8} = 0.285 \mu\text{m}^2$. Similarly, we can compute the scaling of the logic. Approximate metric for density scaling is the product of contacted poly pitch (CPP) and minimum metal pitch (MXP). In TSMC process $CPP \cdot MXP = 0.16 \cdot 0.18 \mu\text{m}^2$ at 65 nm and $CPP \cdot MXP = 0.09 \cdot 0.064 \mu\text{m}^2$ at 16 nm, leading to a scaling factor $\frac{0.16 \cdot 0.18}{0.09 \cdot 0.064} = 5$. Nevertheless, improvements in cell library design for advanced technology nodes can further improve this scaling ratio. For example, TSMC reports us to 2.35x density improvement from 65 to 40 nm, 2x improvements from 40 to 22 nm, and 1.44x improvements from 22nm to 16 nm, yielding to a cumulative logic density improvement of 6.8x from 65 to 16 nm. For considering a best-case scenario for designing the SRAM logic at 16 nm we consider an logic density scaling factor of 6.8x. Figure R2 shows the layout at 65 nm of the SRAM based accelerator in the reference¹⁸. We can directly measure the memory area (bit cell array) and logic (CTRL), which at 65 nm results in a total of $A_{SRAM65} = 0.56 \text{ mm}^2$. By applying the aforementioned scaling rules we can compute the area at 16 nm as $A_{SRAM16} = \frac{A_{SRAM65}}{6.8} = 0.082 \text{ mm}^2$. In the case of

SRAM accelerator, the maximum performance is $\tau_{SRAM} = 2.89 \text{ GOPS}^{18}$ which leads to an area efficiency of

$$\alpha_{SRAM} = \frac{2.89 \text{ GOPS}}{0.082 \text{ mm}^2} = 35.24 \frac{\text{GOPS}}{W}$$

While the area occupation for SRAM based accelerator is significantly lower compared to analog CAM accelerator, the resulting area efficiency is also drastically lower. This is due to the full parallelism that it is possible to achieve thanks to analog CAM non volatile and multi bit behaviour. We can thus conclude that the area efficiency of analog CAM compared to SRAM is $\sim 160x$ better when accelerating RF inference.

We added a comparison in Supplementary Materials.

4. The detailed neural network structure has been not presented in this paper. Considering that hardware architecture such as memory size and data mapping are determined by neural network structure, it is necessary to show the detailed neural network.

We thank the reviewer for the comment. Details of the random forest model are given in SI fig4 and SI fig5. The neural network autoencoder is used for compression purpose in Fig. 4 g-e-f and the structure is illustrated there. To clarify we added a figure in the Supplementary Materials showing the 32x32 input and the corresponding compressed latent space which is used for training the RF classification.

5. In table 1, the proposed ML accelerator is designed using 16nm CMOS technology while the previous works used 40nm~65nm CMOS technology. Those accelerators should be compared under same basis.

We thank the reviewer for the comment. Based on the reviewer suggestion we updated Table 1 with the calculation of power, throughput and energy consumption based on 65 nm technology showing an improvement in throughput and energy consumption of 900x and 1.5x, respectively. Note that the throughput of analog CAM accelerator stays constants at constant clock even with different technologies since it is possible to modulate the conductance range of memristors based on the parasitic capacitance that needs to charge/discharge (as an example for 16nm technology implementation a conductance range 10nS-2uS has been used, while 125nS-25uS has been used for implementing the same RF accelerator with 65 nm technology).

We updated the text accordingly.

1. Ielmini, D. & Wong, H.-S. P. In-memory computing with resistive switching devices. *Nat Electron* **1**, 333–343 (2018).
2. Li, C. *et al.* Analog content-addressable memories with memristors. *Nat Commun* **11**, 1638 (2020).
3. Gunning, D. Explainable artificial intelligence (xai). <https://www.darpa.mil/program/explainable-artificial-intelligence> (2017).
4. Kaggle. State of machine learning and data science 2020. <https://www.kaggle.com/kaggle-survey-2020> (2020).
5. Guillaume-Bert, M., Bruch, S., Gordon, J. & Pfeifer, J. Introducing TensorFlow Decision Forests. <https://blog.tensorflow.org/2021/05/introducing-tensorflow-decision-forests.html> (2021).
6. Lundberg, S. M. *et al.* From local explanations to global understanding with explainable AI for trees. *Nat Mach Intell* **2**, 56–67 (2020).
7. Lundberg, S. M. *et al.* Explainable machine-learning predictions for the prevention of hypoxaemia during surgery. *Nature Biomedical Engineering* **2**, 749–760 (2018).
8. Yan, L. *et al.* An interpretable mortality prediction model for COVID-19 patients. *Nat Mach Intell* **2**, 283–288 (2020).
9. Graves, C. E. *et al.* In-Memory Computing with Memristor Content Addressable Memories for Pattern Matching. *Adv. Mater.* **32**, 2003437 (2020).
10. Ni, K. *et al.* Ferroelectric ternary content-addressable memory for one-shot learning. *Nat Electron* **2**, 521–529 (2019).
11. Hu, M. *et al.* Dot-product engine for neuromorphic computing: programming 1T1M crossbar to accelerate matrix-vector multiplication. in *Proceedings of the 53rd Annual Design Automation Conference on - DAC '16* 1–6 (ACM Press, 2016). doi:10.1145/2897937.2898010.
12. Li, C. *et al.* Efficient and self-adaptive in-situ learning in multilayer memristor neural networks. *Nat Commun* **9**, 2385 (2018).

13. Ambrogio, S. *et al.* Equivalent-accuracy accelerated neural-network training using analogue memory. *Nature* **558**, 60–67 (2018).
14. Le Gallo, M. *et al.* Mixed-precision in-memory computing. *Nat Electron* **1**, 246–253 (2018).
15. Li, C. *et al.* Analogue signal and image processing with large memristor crossbars. *Nat Electron* **1**, 52–59 (2018).
16. Sheridan, P. M. *et al.* Sparse coding with memristor networks. *Nature Nanotech* **12**, 784–789 (2017).
17. Zidan, M. A. *et al.* A general memristor-based partial differential equation solver. *Nat Electron* **1**, 411–420 (2018).
18. Kang, M., Gonugondla, S. K., Lim, S. & Shanbhag, N. R. A 19.4-nJ/Decision, 364-K Decisions/s, In-Memory Random Forest Multi-Class Inference Accelerator. *IEEE J. Solid-State Circuits* **53**, 2126–2135 (2018).
19. Burr, G. W., Lim, S., Murmann, B., Venkatesan, R. & Verhelst, M. Fair and comprehensive benchmarking of machine learning processing chips. *IEEE Des. Test* 1–1 (2021)
doi:10.1109/MDAT.2021.3063366.
20. Sheng, X. *et al.* Low-Conductance and Multilevel CMOS-Integrated Nanoscale Oxide Memristors. *Adv. Electron. Mater.* **5**, 1800876 (2019).

Reviewers' Comments:

Reviewer #1:

Remarks to the Author:

This reviewer appreciates the detailed responses provided by the authors, and the resulting modifications made to the manuscript. I do not have any additional comments.

Reviewer #2:

Remarks to the Author:

The authors have addressed my comments, and the paper is good to publish. The authors better improve the quality of Fig.s 2a and 4a.

Reviewer #3:

Remarks to the Author:

1.Despite of the author's response, this reviewer still thinks that the novelty of this paper is not enough because it simply combines analog CAM and tree-based machine learning that have been previously studied [36]. Also, it is not clear why the newly proposed computing concept has advantages in terms of energy and area efficiency in comparison to crosspoint based computing. This paper does not include the difference from crosspoint based machine learning computing.

2.In the decision of the cam array size, the author provides throughput and individual power consumptions in each component of the proposed memory to achieve an optimal energy efficiency. However, this reviewer thinks that providing the TOPS/W versus various array size is more straightforward to understand the design decision of this work. Furthermore, as shown in Figure 11 of supplementary material, the percentage of power consumed by the analog CAM varies at different technologies. In this case, it is not clear whether the CAM array size will remain the same or change. Therefore, the author should provide more clear analysis to support their design decision.

We thank the editor and reviewers for their careful review of our work. Based on these comments and suggestions we have revised the manuscript. All changes to the text are highlighted in the revision and we have provided point by point responses below. The main changes made to the manuscript are:

- a) We improved the main text based on Reviewer 3 suggestions
- b) We improved the quality of all figures

Reviewer 1

This reviewer appreciates the detailed responses provided by the authors, and the resulting modifications made to the manuscript. I do not have any additional comments. We thank the reviewer for the positive comment on our revised work and for the previously given inputs that improved the quality of our manuscript.

Reviewer 2

The authors have addressed my comments, and the paper is good to publish. The authors better improve the quality of Fig.s 2a and 4a.

We thank the reviewer for carefully reading the revised manuscript and for the previously provided comments which increased the quality of our work. We increased the quality of all Figures by using a vectorial format.

Reviewer 3

We thank the reviewer for carefully reading our work and providing comments to increase our work quality. We provided a point by point response below.

1. Despite of the author's response, this reviewer still thinks that the novelty of this paper is not enough because it simply combines analog CAM and tree-based machine learning that have been previously studied [36]. Also, it is not clear why the newly proposed computing concept has advantages in terms of energy and area efficiency in comparison to crosspoint based computing. This paper does not include the difference from crosspoint based machine learning computing.

We thank the reviewer for the comment. Implementing tree-based machine learning with in-memory computing with memristor has not been shown yet neither with crosspoint arrays nor CAM. For this reason it is not possible to perform an “apples-to-apples” comparison of energy efficiency and area on the same computing workload with crosspoint arrays. We hope that the publication of our work will now allow such a comparison in the future.

We revised the text to clarify these points:

2. In the decision of the cam array size, the author provides throughput and individual power consumptions in each component of the proposed memory to achieve an optimal energy efficiency. However, this reviewer thinks that providing the TOPS/W versus various array size is more straightforward to understand the design decision of this work.

We thank the reviewer for the comment. Figure R1 shows the energy efficiency in TOPS/W as function of width and height of the analog CAM array:

Figure R1 Energy efficiency as function of width and height of the analog CAM arrays expressed in TOPS/W without (a) and with (b) pipelined architecture.

Fig. R1(a) is in agreement with Fig. 5f where the optimum is obtained for large and tall arrays. However this calculation does not include the pipelined architecture, where after using a tile corresponding to a given set of feature vector it can be freed and fed with new data, assuming a constant data stream from the input. Fig. R1 (b) shows the efficiency as function of width and height for a pipelined architecture highlighting the benefit of thin arrays (small W). The latter is in agreement with Supplementary Information Figure 9, which shows the energy per decision in the pipelined architecture. Thus TOPS/W calculation shows the same results as energy per decision, since

$$\frac{TOPS}{W} \sim \frac{1}{Energy/dec}$$

As pointed out in the previous revision, TOPS and TOPS/W are not ideal metrics for system level benchmarking of tree based machine learning for analog CAM implementation. We also believe that

adding other studies of performance as function of array size could be misleading in what is the actual value to optimize, which for this application is the throughput and energy per decision per node.

We added a sentence in the main text to clarify why we chose the dimension of $H = 480$ and $W = 16$.

3. Furthermore, as shown in Figure 11 of supplementary material, the percentage of power consumed by the analog CAM varies at different technologies. In this case, it is not clear whether the CAM array size will remain the same or change. Therefore, the author should provide more clear analysis to support their design decision.

We thank the reviewer for the comment. Indeed the percentage of power consumed by the analog CAM varies at different technologies. The analog CAM cells scale down with the technology node, but transistors T1 and T3 are kept large in order to allow for enough current during the programming operation. For this reason area/power contribution of the analog CAM array compared with the peripherals becomes dominant. However, for simplicity the array size (H and W) is considered constant.

We revised the Supplementary Information Fig. 10 label to clarify these points.